# Technology empowerment: Digital transformation and enterprise ESG performance—Evidence from China's manufacturing sector

Xianyun Wu, Longji Li *, Dekuan Liu, Qian Li

School of Management, Dalian Polytechnic University, Dalian, China

* m1446050591@163.com

## Abstract

In light of the long-term constraints posed by the "dual carbon" objective, can digital technology emerge as a transformative solution for enterprises to embark on a sustainable development trajectory? The existing body of research has yet to reach a consensus. In order to shed further light on the intricate relationship between digital transformation and ESG performance of enterprises, this study empirically examines the mechanisms and boundaries through which digital transformation influences ESG performance, based on observational data from A-share manufacturing listed companies in Shanghai Stock Exchange and Shenzhen Stock Exchange spanning from 2011 to 2021. The findings demonstrate that digital transformation exerts a significant positive impact on the ESG performance of manufacturing enterprises. Mechanism analysis reveals that the enabling effect of digital transformation primarily enhances company transparency, thereby fostering continuous improvements in ESG performance among manufacturing enterprises. The performance expectation gap will give rise to the phenomenon of "stop-loss in time" and impede the promotional impact of digital transformation. Further investigation into industrial characteristics and industry competition intensity indicates that state-owned enterprises and those operating within highly competitive environments experience more pronounced effects of digital transformation on their ESG performance. This study expands the mechanism and boundary of digital transformation on ESG performance of manufacturing enterprises, and provides a new perspective for manufacturing enterprises to realize the collaborative transformation of digital and green.

## Introduction

In 2004, the United Nations introduced the concept of ESG (Environmental, Social, and Governance) in its initiative report titled "Who Cares Wins" [1]. This report provided a new direction for businesses on how to implement sustainable development principles. The concept of ESG originates from ethical investment and responsible investment, rejecting the profit-

**Funding:** This study is a stage research result of the Liaoning Economic and Social Development Research Project 2024 (Project No. 2024lslybwzzkt-034), the Liaoning Social Science Planning Fund Educational Science Project (Project No. L21AED005), and the Dalian Municipal Science and Technology Bureau's Soft Science Project (Project No. 2023JJ13FG087).

**Competing interests:** The authors have declared that no competing interests exist.

centric business philosophy, and advocating for enterprises to incorporate environmental, social, and governance factors into their investment decisions while considering economic benefits [2–4]. Currently, there is a global wave of low-carbon transformation underway, leading all countries worldwide to introduce ESG-related policies and regulations. Examples include "the Corporate Sustainability Reporting Directive" and "IFRS S1—General Requirements for Disclosure of Information Sustainability-related Financial Information". In recent years, China's "dual carbon" goal has accelerated the development process of ESG in China [5]. Regulators have issued a series of policies and regulations that gradually require listed companies to disclose ESG-related information; thus making ESG practices an essential aspect for enterprise development. However, challenges such as insufficient willingness and limited participation in specific corporate practices undermine the positive impact of the ESG system on China's economic transformation. Therefore, it is crucial to explore both internal and external factors influencing enterprises' performance in implementing ESG.

At present, China is in a critical period of transformation from a manufacturing power to a manufacturing power [6]. Manufacturing is the backbone of the country's economic development, Facing the medium and long term constraints of "dual carbon" target, whether manufacturing enterprises can explore a sustainable transformation path is related to the long-term healthy development of China's economy [7]. The wave of digital transformation offers a novel perspective for the sustainable development of manufacturing enterprises. Digital transformation is regarded as the extensive application of digital technology across various aspects of enterprise survival, operation, and sales [8]. Previous studies have demonstrated that the adoption of digital technology can enhance the economic efficiency of manufacturing enterprises by improving resource allocation efficiency, innovation capability and Profit level [9–11]. However, can the technological advancements and resource utilization resulting from digital transformation effectively stimulate the inherent capabilities of manufacturing enterprises to enhance their environmental, social, and governance (ESG) performance? Although previous studies have made preliminary explorations into the relationship between digital transformation and ESG performance [12, 13], the mechanism underlying digital transformation remains incompletely elucidated, necessitating further exploration of working conditions. Therefore, this study aims to further expand the existing research on this topic in order to address the limitations identified in previous studies.

Building upon China's "dual carbon" goal policy context, this study delves into the potential of digital transformation in the manufacturing industry to stimulate endogenous drivers for enhancing ESG performance within enterprises. This investigation aims to unveil the underlying mechanisms of digital transformation, augment existing research findings, and hold significant theoretical and practical implications. Consequently, this study adopts corporate transparency as a foundational aspect and integrates the performance expectation gap into its research framework. Empirical analysis is conducted using observation data from A-share manufacturing listed companies on Shanghai and Shenzhen Stock Exchanges spanning from 2011 to 2021 to examine the boundaries and mechanisms through which digital transformation influences corporate ESG performance.

Compared to previous studies, this study innovatively addresses the following aspects: (1) Previous studies did not investigate whether the relationship between digital transformation and ESG performance of enterprises would be influenced during periods of declining enterprise performance. By introducing the situational condition of performance period gap, this study further defines the impact of digital transformation on ESG performance and enriches research on between digital transformation and performance feedback. (2) From a corporate transparency perspective, this paper elucidates the mechanism through which digital

transformation affects ESG performance in manufacturing enterprises, offering new theoretical references and practical insights for sustainable development enabled by digital technology.

## Literature review

Since the inception of the ESG concept in 2004, it has garnered significant attention from investors and business managers owing to its unique ability to balance economic benefits with social values. Consequently, academic research in ESG-related fields has witnessed substantial growth [14], with scholars predominantly favoring investigations into the impact of ESG [15]. Mainstream scholars contend that ESG practices can enhance enterprise brand valuation and foster green innovation capabilities, thereby mitigating business risks and ultimately improving enterprise value [16–19]. Scholars have also started examining the influencing factors of enterprise ESG performance. Previous research indicates that factors such as regional digital finance development and environmental protection tax legislation can significantly contribute to enhancing enterprise ESG performance [20, 21]. However, existing studies pay more attention to the external factors that affect the ESG performance of enterprises. In order to fully play the positive role of ESG system in the low-carbon transformation of Chinese enterprises, it is necessary to stimulate the endogenous motivation of enterprises to improve ESG performance.

With the advent of a new wave of scientific and technological revolution, digital technologies such as big data and blockchain offer a novel avenue for facilitating the high-quality development of manufacturing enterprises. Esteemed scholars contend that leveraging digital technology can enhance resource allocation efficiency, innovation capabilities, and customer information advantage, thereby fostering the high-quality development of manufacturing enterprises [9, 11, 22]. In addition to researching the economic benefits of digital transformation, scholars have also begun to focus on its non-economic value. Specifically, they argue that the application of digital technology can facilitate green innovation in enterprises and lead to a reduction in carbon emissions [23, 24]. With the advancement of research, scholars have started to establish a connection between digital transformation and enterprise ESG performance, leading to two main categories in existing research findings: the "empowerment" effect and the "too much is not good" effect. The "empowerment" effect is specifically reflected in the fact that digital transformation can improve the ESG performance of enterprises by reducing agency costs and improving corporate reputation and dynamic capabilities [25, 26]. The "too much is not good" effect is specifically reflected in the fact that a high level of digitalization may weaken the ability and motivation of enterprises to carry out ESG practices. Asymmetric digital transformation and organizational transformation process make it difficult to play the enabling effect of digital technology, which may lead to "information overload" and reduce the information processing ability of enterprises. In addition, a large amount of capital investment in the materialization of digital technology may induce "crowding-out effect" and delay the process of enterprise green transformation [27–29].

The concept of transparency emerged from research in the field of information disclosure [30]. As research on information disclosure expanded, scholars introduced the notion of "company transparency," which refers to providing specific company information to external stakeholders [31]. With the deepening of research, Chinese scholars have refined the concept of company transparency, that is, the higher the transparency of a company, the wider and deeper the scope and level of external investors' access to internal information of a company, and the stronger the liquidity of information [32]. The application of digital technology offers a novel perspective for researching company transparency. However, upon reviewing existing literature, it is evident that scholars tend to associate digital transformation with analysts'

forecasts and corporate governance [33, 34]. These studies suggest that while there may be a close relationship between digital transformation and company transparency, further exploration is necessary.

The aforementioned analysis reveals that despite the existence of relevant studies demonstrating the correlation between digital transformation and ESG performance, certain limitations persist, primarily in the following aspects: (1) the existing research mainly discusses the influence between the two from the perspective of internal control, green innovation and information disclosure quality, and its internal influence mechanism needs to be further expanded. (2) The measurement approach for assessing the extent of digital transformation within enterprises remains singular, making it challenging to mitigate potential deviations resulting from false corporate disclosures. (3) What are the requisite conditions for effectively harnessing the impact of digital transformation empowerment?

Building upon this premise, the present study adopts corporate transparency as a focal point, integrates the performance expectation gap into the research framework, and explores whether digital transformation can incentivize enterprises to engage in ESG practices. The present study contributes to the existing literature on the mechanisms of digital transformation, elucidates the impact of digital transformation in situations characterized by performance expectation gaps, and addresses a research gap in this domain.

## Theoretical analysis and research hypotheses

### Digital transformation and enterprise ESG performance

The process of digital transformation involves a comprehensive reshaping of the traditional business model, governance mechanism, and organizational structure of an enterprise by integrating artificial intelligence, big data, blockchain, and other digital technologies into various aspects such as production, sales, and transportation [35]. Existing literature primarily focuses on the economic performance of digital transformation and its individual non-economic aspects [36, 37], while only recently has there been exploration of the relationship between digital transformation and integrated environmental, social, and corporate governance (ESG) performance [38]. The present study posits that the digital transformation is poised to enhance the ESG performance of manufacturing enterprises through bolstering their capabilities and fostering intrinsic motivation.

From the perspective of behavioral outcomes, digital transformation improves the comprehensive strength of enterprises to carry out ESG practices. First of all, the rapid development of digital finance has broadened the financing channels of manufacturing enterprises [39]. It has also improved the matching efficiency of both parties of credit, effectively reduced the probability of resource mismatch and credit default, solved the financial discrimination problem of "Large enterprises are allocated a substantial loan quota, whereas small enterprises receive a limited loan quota" [40]. To a certain extent, and reduced the dependence of manufacturing enterprises on "resource-based" shareholders and large customers due to financing constraints [9, 41]. This has greatly improved the discourse power of environment-sensitive executives and improved the intellectual support for enterprises' ESG practices. In addition, the application of digital technology can refine the production and research and development process of products, reduce the probability of research and development manipulation [42], and provide conditions for enterprises to give full play to green innovation resources. This undoubtedly helps improve the green innovation ability of manufacturing enterprises, and then promote the quality and efficiency of green patents of enterprises, and provide technical support for ESG practices of manufacturing enterprises [43, 44]. The application of digital platforms and big data technology has broken the barriers to information acquisition of manufacturing

enterprises, narrowed the distance between enterprises and customers, and enabled enterprises to accurately grasp the differentiated needs of customers and improve the competitiveness of enterprises' products [45]. At the same time, the application of digital technology improves the ability of enterprises to integrate internal resources and acquire external resources, blurs the business boundary of enterprises, transforms the single chain management structure of enterprises into a diversified network management structure, improves the sustainable competitiveness of enterprises, and provides economic possibilities for enterprises' ESG practices [46, 47]. Digital transformation consolidates the overall strength of manufacturing enterprises through the three aspects of "talent-technology-economy", and provides realistic conditions for manufacturing enterprises to improve their ESG performance.

From the perspective of behavioral motivation, digital transformation improves the willingness of manufacturing enterprises to carry out ESG practices. On the one hand, the application of digital technology breaks the constraints of time and space of traditional information exchange, connects stakeholders together through digital platforms, and improves the frequency of internal and external information interaction of enterprises [48]. Active disclosure of enterprises is no longer the only channel for stakeholders to obtain enterprise information, narrowing the "information fault line" between enterprises and stakeholders. It provides an opportunity for external investors to realize the identity transformation from "free rider" to "administrator" in corporate governance [49]. In addition, the application of big data technology makes any behavior of enterprises to follow, and R&D manipulation, false information disclosure and other violations are contained, which promotes the improvement of the quality of information disclosed externally and strengthens the internal motivation of enterprises to improve ESG performance [50, 51]. On the other hand, digital transformation, as a positive signal of change, will attract the attention of external market players such as the government, analysts and media [52]. When enterprises are placed under the "spotlight", their business behaviors will be amplified infinitely, resulting in a sharp increase in the pressure of external attention, which is both an opportunity and a challenge for enterprises. Positive ESG practices will be spread rapidly by the media and analysts, improve the corporate image, and gradually increase its recognition among the government and consumers, It improves the advantages of enterprises in obtaining political resources and consumer trust [4, 53]. However, when market observers dramatize poor market performance, the negative impact of enterprises rises geometrically and may be "labeled" as a shackles that restrict the development of enterprises. Therefore, in this case, the willingness of enterprises to ESG practices will increase significantly. Based on the above analysis, this paper proposes the following hypothesis:

H1: Digital transformation will promote the improvement of ESG performance of manufacturing enterprises.

## Digital transformation, company transparency and corporate ESG performance

This study posits that digital transformation primarily enhances company transparency, thereby continuously improving the ESG performance of manufacturing enterprises. On the one hand, it mitigates the issue of information asymmetry and facilitates external shareholders' participation in corporate governance through its traceability, immutability, and timeliness [54, 55]. Furthermore, it curbs managers from exploiting information asymmetry to manipulate environmental and social responsibility for profit-driven stock price escalation while enhancing internal governance transparency to improve non-financial information disclosure quality [50].

On the other hand, the application of digital technology will improve the circulation frequency of internal and external information of enterprises. The existence of asymmetric information between enterprises and stakeholders also gives rise to stakeholders' distrust and even aversion to enterprises with high information acquisition costs and low information disclosure quality, which reduces the market attention of such enterprises [56, 57]. In order to obtain more external resources to make up for the loss of sustainable strategy, enterprises are more willing to take advantage of the convenience of digital technology and the characteristics of low information disclosure cost to actively promote the positive achievements of corporate environment and social responsibility, and shift from passively improving the quality of information disclosure to actively improving it [58]. At the same time, the diversified information sharing channels derived from digital transformation make it easier for enterprises to identify false or low-quality information disclosure behaviors, which improves the quantity, quality and depth of enterprise information obtained by stakeholders [34]. External analysts, media and other market intermediaries can make more objective and fair market evaluations [59, 60]. It helps enterprises to improve their green and environmental reputation among consumers and governments, and encourages enterprises to carry out ESG practices with confidence. To sum up, company transparency is the channel through which digital transformation can improve the ESG performance of manufacturing enterprises. Based on this, this paper puts forward the following hypothesis:

H2: The digital transformation facilitates the enhancement of ESG performance through augmenting company transparency.

## Digital transformation, performance expectation gap and enterprise ESG performance

The performance expectation gap refers to the difference between an enterprise's actual performance and its expected performance [61]. According to the theory of corporate behavior, the performance expectation gap is an important reference for managers to formulate corporate future strategies [62]. Among them, the expected performance represents the minimum level of output anticipated by management, and whether the actual performance aligns with management's expectations will significantly influence subsequent strategic planning decisions.

Currently, there is no consensus among academia regarding the potential impact of the performance expectation gap. On one hand, when managers observe that actual performance falls short of expectations, it may lead to a "make or break" situation. According to the Resource Based View, an enterprise's competitive advantage relies on its unique resources [63]. When an enterprise fails to meet expectations in terms of performance, its competitive advantage begins to decline. As a crucial component of enterprises' sustainable development strategy, ESG practices may temporarily compromise their operational performance due to high investment costs and extended return periods. However, forward-thinking managers recognize that ESG practices hold significant appeal in terms of corporate reputation, political resources, and consumer recognition [64, 65]. In order to establish sustainable competitive advantages for enterprises, managers are more inclined to forego short-term interests and pursue long-term developmental benefits. At the same time, the talent reserve and organizational structure of enterprises need to be timely matched with the process of digital transformation to play an enabling role [66]. However, because the enterprise performance is not up to expectations, the capital market will cause doubts about the operating conditions of enterprises, making it more difficult for enterprises to obtain resources from the outside. In the face of "internal and external challenges", managers will use limited organizational resources to make up for the gap

between the application of digital technology and the organizational governance system, give full play to the enabling role of digital technology, and improve the level of digital governance of enterprises [30].

However, the performance expectation gap can also result in the occurrence of a phenomenon known as "stop-loss in time"effect. The decline in business performance leads to internal anxiety among management and doubts from external investors, which subsequently affects managers' judgment and execution capabilities [67, 68]. Since both digital transformation and ESG practices require significant resource investments, companies that are struggling financially may find it challenging to sustain these high-cost reform solely with their own resources. As a result, the enterprise's transformation process slows down and ESG practices are reduced or even suspended. Additionally, according to threat-rigidity theory [69], when faced with continuous expectation gaps, enterprises tend to prioritize survival over thriving. Consequently, decision-making becomes more conservative as organizations immersed in pessimism experience sluggish information processing and acceptance capacities [70, 71]. In such circumstances, limited market information becomes the basis for strategic decisions made by management. Choosing riskier reforms or investments during this period would expose enterprises to devastating strategic risks that not only fail to alleviate their predicament but also deplete their resources further. Furthermore, lack of resources exacerbates the difficulty of implementing ESG practices at this time. Even if enterprise management is willing to exhaust all options in pursuing original strategic goals, they remain powerless due to resource constraints [72]. Based on the aforementioned analysis, this paper proposes the following hypothesis:

H3a: Performance expectation gap has a positive moderating effect on the relationship between digital transformation and ESG performance of manufacturing enterprises.

H3b: Performance expectation gap has a negative moderating effect on the relationship between digital transformation and ESG performance of manufacturing enterprises.

## Research design

### Research sample and data sources

This study utilizes a sample of manufacturing enterprises listed on the A-shares of the Shanghai Stock Exchange and Shenzhen Stock Exchange, covering the period from 2011 to 2021. To ensure consistency with previous studies, the initial sample is refined through the following steps: ① Exclusion of samples classified as ST and *ST in the current year; ② Elimination of samples with missing data on core variables; ③ Exclusion of samples with less than three consecutive years of data; ④ To mitigate the impact of extreme values, all continuous variables are winsorized at the 1% and 99% levels. Consequently, a total of 6044 observation samples are obtained.

The original financial data utilized in this study, as well as the robustness test concerning the extent of digital transformation, were sourced exclusively from the China Stock Market & Accounting Research Database(CSMAR). Furthermore, the word frequency analysis pertaining to digital transformation primarily relied upon annual reports disclosed by listed companies through Juchao Consulting Network, Shenzhen Stock Exchange, and Shanghai Stock Exchange. The data analysis was conducted using Python and Stata version 16.0.

### Measurement of variables

**Dependent variable.** ESG performance (ESG). Currently, there exist notable disparities in the measurement of ESG ratings domestically and internationally, with influential rating

systems including MSCI, Bloomberg, Shangdao Ronglv, Huazheng, among others. The ESG rating score provided by Bloomberg was chosen as the proxy index for the core explanatory variable, based on the sample characteristics outlined in this paper.

**Independent variables.** Digital transformation (Digital). The measurement method employed in this study for assessing the extent of digital transformation primarily draws upon existing research [73], utilizing the construction of digital dictionaries and text analysis to determine the degree of digital transformation within enterprises. In contrast to previous approaches that relied on intangible assets related to digital technology, questionnaire surveys, and ERP system applications [74–76], this measurement method establishes a relatively objective and comprehensive digital term dictionary based on semantic expressions found in national policies pertaining to the digital economy. Subsequently, it employs text analysis techniques to construct a more holistic indicator reflecting the level of digitization among Chinese enterprises. Meanwhile, considering the "right-skewed" feature word frequency data and avoiding the impact of enterprises not carrying out digital transformation, the total word frequency is added by 1 and then logarithmized.

**Mediating variables.** Company transparency (Tra). Drawing upon the methodologies proposed by LANG et al. (2012) and Xiang et al. (2020) [77, 78], this study adopts four comprehensive indicators to assess company transparency: earnings quality, audit company quality, information disclosure rating, and analyst attention.

The first indicator is earnings quality, and this paper chooses DD model to measure corporate earnings quality:

$$TCA_{i,t} = a_1 + a_2 CFO_{i,t-1} + a_3 CFO_{i,t} + a_4 CFO_{i,t+1} + a_5 \Delta REV_{i,t} + a_6 PPE_{i,t} + \varepsilon_{i,t} \qquad (1)$$

Where TCA represents total current accruals, defined as operating profit minus operating cash flow plus depreciation and amortization expense; CFO denotes operating cash flow; ΔREV signifies change in operating income; PPE refers to the value of fixed assets at year-end. All variables are normalized by dividing them with annual average total assets to mitigate the impact of firm size. The residual value was derived by conducting sub-annual regression analysis. Subsequently, the standard deviation is computed based on the five-year residuals from year t and its preceding four years, thereby obtaining the enterprise's earnings quality index for year t. Additionally, considering comparability with other indicators, the earnings quality index is multiplied by -1.

The second indicator is the quality of the audit company, which is measured by whether the listed company employs the auditors of the Big Four domestic accounting firms to conduct audit.

The third indicator is the information disclosure rating, which primarily pertains to the disclosure ratings of the Shanghai and Shenzhen Stock Exchanges. In this context, A denotes excellent, B represents good, C signifies pass, and D indicates fail. This study assigns a numerical value to each rating in descending order: A = 4 and D = 1. Consequently, a higher score corresponds to a superior quality of information disclosure.

The fourth indicator is analyst attention, referring to the existing research, how many analysts (teams) have followed the company within a year. in order to avoid the impact of 0 value, it is added by 1 to take the logarithm.

Based on the aforementioned four indicators, this study constructs a comprehensive indicator to assess company transparency (Tra) by adopting the approach proposed by Xin Qingquan et al. (2014) [79]. This is accomplished as follows: computing the average of sample percentiles for each variable. Considering the delayed initiation of SSE's information disclosure rating and missing data in certain years, the company transparency index is determined

as the average of three remaining index sample percentiles. A higher Tra index indicates greater company transparency.

**Moderating variables.**   Performance expectation gap, referring to Qiu et al. (2022) [80], is measured by the difference between actual business performance and expected business performance. The specific calculation formula is as follows:

$$HA_{i,t} = \alpha HA_{i,t-1} + (1 - \alpha)P_{i,t-1} \tag{2}$$

$$A_{i,t} = \beta HA_{i,t} + (1 - \beta)\, SA_{i,t} \tag{3}$$

The enterprise's historical expected performance ($HA_{i,t}$) is determined by a weighted combination of its historical expected performance in period t − 1 and the actual operating performance in period t − 1, where α represents the weight assigned to this combination and takes a value between (0,1). Following the practices of Cao Yanan (2023) and Chen(2008) [81, 82], we set α as 0.4 for calculating historical expected performance. The comprehensive expected performance is calculated by weighting the historical expected performance of the enterprise and the industry's expected performance. SA represents the enterprise's expected performance in relation to the industry, which is determined as the mean value of ROA for all enterprises in the industry except Company i. The β weight setting follows Guo Rong et al. (2019) and Rudy (2016) [83, 84]. Initially set at 0.5, β increases by 0.1 incrementally each time. The weight is determined based on model fitting, with results indicating that the best fit occurs when β = 0.5; therefore, this paper selects β = 0.5 to weigh the comprehensive expected performance.

When the actual performance falls below the expected performance (P-A<0), a negative gap is observed between the actual and expected performances. Conversely, when the actual performance exceeds the expected performance, a positive gap in expected performance is evident. To further analyze the impact of digital transformation on ESG performance of manufacturing enterprises considering this expectation-performance gap, we introduce a dummy variable L1 in this study. The value of L1 is set to 1 when the expectation-performance gap is < 0 and 0 when it is ≥0. The constructed variable L1*$gap_{i,t}$ represents instances where actual performance lags behind expectations, with smaller values indicating larger gaps. Additionally, for ease of comprehension, we multiply L1*$gap_{i,t}$ by -1 to obtain an indicator for the expected performance gap ($Ngap_{i,t}$). Higher values indicate greater disparities between actual and anticipated performances.

**Control variables.**   Drawing on the existing literature, This paper adds enterprise Size (Size), asset-liability ratio (Lev), growth rate of operating income (Grow), Cashflow ratio (Cashflow), proportion of independent directors (Indira), years of company establishment (Listage) and shareholding ratio of the largest shareholder (Top1) as control variables in the regression model. The detailed variable definition and calculation method are shown in Table 1.

## Model setting

In terms of setting the benchmark model, we adopt a methodology commonly employed in previous studies [19, 85] and construct the following model to empirically test H1 as proposed in this study:

$$ESG_{i,t} = \beta_0 + \beta_1 Digital_{i,t} + \sum control_{i,t} + Year + Industry + \varepsilon_{i,t} \tag{4}$$

Where i represents the enterprise and t represents the year. ESGi, t represents the ESG rating index of enterprise i in year t, Digitali, t represents the degree of digital transformation at the

**Table 1. Variable definitions.**

| Variable names | Symbols | Variable explanation |
|---|---|---|
| ESG performance | ESG | Bloomberg ESG Ratings Index |
| Digital transformation | Digital | The logarithm of digital transformation word frequency +1 in the annual report |
| Company transparency | Tra | The percentages of earnings quality, analyst attention, audit firm quality, and disclosure rating were averaged |
| Performance expectation gap | Ngap | The difference between actual and expected performance of a firm ≥0 is assigned a value of 0, and < 0 is multiplied by -1 |
| Enterprise size | Size | Take the logarithm of the total assets of the business |
| Asset-liability ratio | Lev | Total year-end responsible/total year-end assets |
| Growth rate of operating income | Grow | Current operating income/previous operating income |
| Cash flow ratio | Cashflow | Net cash flow from operating activities/total assets |
| Percentage of independent directors | Indira | Number of independent directors/total number of board members |
| Number of years since the establishment of the company | Listage | The logarithm of the number of years since the establishment of the enterprise at the end of the current year +1. |
| Shareholding ratio of the largest shareholder | Top1 | Number of shares held by the largest shareholder/total share capital at the end of the year |

enterprise-year level, and Σcontroli,t represents all the control variables in this paper. In addition to the above control variables, this paper controls the industry and time dummy variables in the model.

## Empirical results analysis

### Descriptive statistics and correlation analysis

The descriptive statistical results of the main variables in this paper are presented in Table 2. Among these, the mean value of ESG performance for manufacturing enterprises is 28.199, indicating a moderate level of environmental, social, and governance performance among the sample manufacturing enterprises that are implementing the "dual carbon" target. there is still significant room for improvement. Furthermore, the minimum value observed among the sample enterprises is 11.488, while the maximum value is 56.121, suggesting a substantial disparity in ESG practice input between leaders and followers. The digital variable exhibits a maximum value of 6.544 and a minimum value of 1.386, highlighting considerable variation in digital transformation degrees across sample enterprises. The results of other control variables are basically similar to those of existing studies [28, 29].

**Table 2. Descriptive statistics.**

| Variables | N | Mean | Min | Max | Sd | P50 |
|---|---|---|---|---|---|---|
| ESG | 6044 | 28.199 | 11.488 | 56.121 | 8.988 | 25.923 |
| Digital | 6044 | 3.646 | 1.386 | 6.544 | 1.108 | 3.583 |
| Size | 6044 | 23.02 | 20.42 | 26.16 | 1.212 | 22.923 |
| Lev | 6044 | 0.449 | 0.0569 | 0.866 | 0.192 | 0.458 |
| Grow | 6044 | 0.170 | 0.414 | 2.042 | 0.331 | 0.1197 |
| Cashflow | 6044 | 0.0644 | 0.108 | 0.255 | 0.0676 | 0.0585 |
| Indire | 6044 | 0.375 | 0.333 | 0.571 | 0.0561 | 0.333 |
| Listage | 6044 | 2.912 | 1.792 | 3.497 | 0.323 | 2.944 |
| Top1 | 6044 | 0.361 | 0.0890 | 0.789 | 0.154 | 0.343 |

Additionally, this study conducts a Pearson correlation test on the main variables, and the results demonstrate a significantly positive correlation coefficient between digital transformation and enterprise ESG performance, thereby providing preliminary support for hypothesis 1 proposed in this paper. Furthermore, the selected control variables exhibit a statistically significant correlation with enterprise ESG performance, indicating the reasonable selection of control variables in this study. The average VIF of each variable in the model regression is 1.18, indicating that there is no serious multicollinearity problem in the model.

## Benchmark regression results

The regression results in Table 3 demonstrate the impact of digital transformation on enterprise ESG performance. In Column (1), only industry and year dummy variables are controlled, while other control variables selected in this study are not included. The regression analysis reveals a significantly positive coefficient of 0.304 for Digital at the 1% level, providing preliminary evidence for a positive correlation between digital transformation and ESG

**Table 3. Regression results.**

| VARIABLES | ESG | ESG | Mechanism test | | Moderating effect |
|---|---|---|---|---|---|
| | | | Tra | ESG | ESG |
| Digital | 0.304*** | 0.278*** | 0.009*** | 0.203** | 0.327*** |
| | (0.112) | (0.097) | (0.002) | (0.094) | (0.101) |
| Tra | | | | 7.982*** | |
| | | | | (0.558) | |
| Ngap | | | | | 4.997 |
| | | | | | (5.326) |
| Digital*Ngap | | | | | -3.000** |
| | | | | | (1.281) |
| Size | | 2.422*** | 0.079*** | 1.790*** | 2.376*** |
| | | (0.092) | (0.002) | (0.096) | (0.094) |
| Lev | | -2.887*** | -0.239*** | -0.979* | -2.421*** |
| | | (0.487) | (0.012) | (0.505) | (0.515) |
| Grow | | -0.430* | 0.007 | -0.489** | -0.576** |
| | | (0.248) | (0.007) | (0.242) | (0.253) |
| Cashflow | | 3.137*** | 0.428*** | -0.280 | 2.628** |
| | | (1.183) | (0.030) | (1.183) | (1.215) |
| Indire | | -1.423 | 0.004 | -1.452 | -1.256 |
| | | (1.432) | (0.037) | (1.405) | (1.432) |
| Listage | | 1.195*** | -0.052*** | 1.607*** | 1.183*** |
| | | (0.289) | (0.008) | (0.285) | (0.288) |
| Top1 | | 2.389*** | 0.056*** | 1.944*** | 2.252*** |
| | | (0.587) | (0.014) | (0.565) | (0.596) |
| Constant | 21.202*** | -34.529*** | -1.411*** | -23.265*** | -33.847*** |
| | (1.874) | (3.653) | (0.074) | (4.039) | (3.714) |
| Observations | 6,044 | 6,044 | 6,044 | 6,044 | 6,044 |
| R-squared | 0.498 | 0.575 | 0.311 | 0.591 | 0.576 |
| Industry | YES | YES | YES | YES | YES |
| Year | YES | YES | YES | YES | YES |

Note: Heteroscedasticity robust standard errors in parentheses;

***, ** and * indicate significance at the level of 1%, 5% and 10%.

performance. Building upon these findings, Column (2) incorporates additional control variables identified in this research, resulting in a slight decrease in the coefficient of Digital; however, it remains statistically significant at the 1% level. These results indicate that even after accounting for industry-specific factors, temporal effects, and firm characteristics, digital transformation continues to play a significant role in enhancing enterprise ESG performance, thereby confirming Hypothesis 1 posited in this paper. The aforementioned findings demonstrate that the benefits derived from digital transformation, including technological advancements, resource optimization, and enhanced management capabilities, can serve as an endogenous driving force for enterprises to engage in ESG practices. These advantages not only provide the necessary material foundation but also offer technical support for businesses to effectively implement ESG initiatives.

Moreover, considering the coefficients of control variables, it can be observed that well-established large enterprises with ample cash flow and high ownership concentration exhibit a greater inclination and capability to leverage the benefits derived from digital transformation in order to enhance their ESG performance. Conversely, enterprises with limited revenue capacity and a high debt ratio display a higher degree of reluctance towards allocating scarce resources for ESG practices due to prevailing survival pressures. The findings of this study are in line with the existing body of research [86, 87].

## Robustness and endogeneity test

**Change the measurement method of variables.**   Firstly, in order to enhance stock prices and attract the attention of uninformed investors, some enterprises tend to embellish facts in their annual reports, extensively publicize their digital transformation blueprint through verbose narratives, and captivate investors with enticing stories and aspirations. Consequently, relying solely on keyword frequency analysis within the annual report becomes inadequate for accurately assessing the extent of digital transformation within these enterprises [88]. Therefore, this study adopts the Digital Transformation Index from CSMAR as a substitute variable that encompasses various dimensions including word frequency related to enterprise transformation, investment in digital resources, formulation of digital strategies, alignment of organizational structure with digital transformation goals, accomplishments in digital transformation endeavors, and application of digital technologies. This comprehensive measurement system aims to rectify the limitations associated with single-indicator assessments. Following a baseline regression approach, we incorporate the Digital Transformation Index (Digital_index) into our model for re-regression analysis. The results are presented as M1 in Table 4 where it is evident that the regression coefficient for Digital_index exhibits significant positive association at a 1% level of significance–consistent with previous findings.

Moreover, this study employs the ESG rating provided by huazheng as an alternative index (ESG_H). Specifically, a value of 9 is assigned to AAA and subsequently decreases in descending order. A higher score indicates better ESG performance of the company. To minimize result deviation caused by different rating systems, the mean value of quarterly ratings is selected as a substitute variable in this paper. The aforementioned empirical method is employed to test the robustness of the baseline regression results, which are presented in M2 within Table 4. Notably, the Digital regression coefficient exhibits significant positive association at a 1% level, thereby further confirming H1 posited in this study.

**Extend the observation period.**   Considering that it takes a certain amount of time for the technological, management and resource advantages brought by digital transformation to affect the ESG practice activities of enterprises, we draw on the practice of existing research to extend the observation period and delay ESG by one, two and three periods. The results are

**Table 4. Robustness test.**

| VARIABLES | M1 ESG | M2 ESG_H | M3 ESG(-1) | M4 ESG(-2) | M5 ESG(-3) | M6 ESG | M7 ESG |
|---|---|---|---|---|---|---|---|
| Digital | | 0.080*** | 0.289*** | 0.254** | 0.209* | 0.542*** | 0.388*** |
| | | (0.016) | (0.105) | (0.110) | (0.116) | (0.148) | (0.131) |
| Digital_index | 0.079*** | | | | | | |
| | (0.011) | | | | | | |
| Size | 2.316*** | 0.249*** | 2.242*** | 2.063*** | 1.912*** | 1.895*** | 2.331*** |
| | (0.092) | (0.013) | (0.100) | (0.105) | (0.109) | (0.189) | (0.129) |
| Lev | -2.882*** | -1.117*** | -2.516*** | -2.023*** | -1.765*** | -3.331*** | -2.625*** |
| | (0.486) | (0.088) | (0.514) | (0.536) | (0.559) | (0.718) | (0.690) |
| Grow | -0.389 | -0.178*** | -0.458* | -0.406 | -0.197 | -0.336* | -0.540* |
| | (0.245) | (0.040) | (0.274) | (0.291) | (0.316) | (0.192) | (0.326) |
| Cashflow | 3.134*** | 0.659*** | 3.446*** | 3.472*** | 2.105 | 1.508 | 4.256*** |
| | (1.180) | (0.205) | (1.233) | (1.308) | (1.368) | (1.132) | (1.605) |
| Indire | -1.732 | 1.403*** | -2.190 | -2.368 | -3.342** | 0.953 | -1.099 |
| | (1.423) | (0.232) | (1.501) | (1.570) | (1.676) | (1.765) | (1.992) |
| Listage | 1.260*** | 0.065 | 1.385*** | 1.602*** | 1.833*** | 2.114 | 1.079*** |
| | (0.282) | (0.048) | (0.308) | (0.334) | (0.371) | (1.404) | (0.409) |
| Top1 | 2.638*** | 0.121 | 2.579*** | 2.596*** | 2.790*** | 1.862 | 2.069** |
| | (0.586) | (0.092) | (0.625) | (0.668) | (0.722) | (1.144) | (0.843) |
| Constant | -33.625*** | -2.710* | -31.622*** | -18.708*** | -27.416*** | -23.092*** | -30.249*** |
| | (3.498) | (1.435) | (2.120) | (2.419) | (3.016) | (5.589) | (2.969) |
| Observations | 6,044 | 6,044 | 5,314 | 4,654 | 3,993 | 5,962 | 3,182 |
| R-squared | 0.578 | 0.135 | 0.528 | 0.509 | 0.484 | 0.848 | 0.551 |
| Industry | YES | YES | YES | YES | YES | NO | YES |
| year | YES | YES | YES | YES | YES | NO | YES |
| High latitude fixed effects | NO | NO | NO | NO | NO | YES | NO |

Note:

***, ** and * indicate significance at the 1%, 5% and 10% levels.

shown in M3, M4 and M5 in Table 4. The regression results are significantly positive at the levels of 1%, 5% and 10% respectively, indicating that H1 in this paper is still robust after the observation period is extended.

**Fixed effect model.** Considering the possible estimation errors caused by unobservable factors that do not change with individuals, this study incorporates individual fixed effects and industry-year joint fixed effects into the model to enhance its robustness. Moreover, given that the fixed effect of high latitude already encompasses the impact of industry and year, dummy variables for industry and year are not included in the regression analysis. The results, presented in M6 of Table 4, exhibit a significantly positive association at a 1% significance level.

**PSM.** In order to address the endogeneity problem arising from potential sample self-selection, this study employs propensity score matching (PSM) for testing purposes. Firstly, enterprises are categorized based on the median degree of digital transformation. Subsequently, all control variables selected in this study are utilized as covariates to pair the samples using a 1:1 nearest neighbor matching method. To ensure the validity of the matching results, a balance test is conducted on the matched outcomes, with all normalized bias absolute values being less than 10%. This indicates that the matching results largely meet the requirements.

Finally, after matching, regression analysis is performed on 3182 samples and presented in M7 of Table 4. The regression coefficient of Digital remains significantly positive at a level of significance of 1%, indicating that the conclusion remains robust after addressing sample self-selection issues.

**Tool variable method.**   Given the potential reverse causality between digital transformation and ESG performance of enterprises, wherein digital transformation can foster improvements in ESG performance while enterprises exhibiting good ESG performance may also demonstrate a greater inclination towards undertaking digital transformation, this study employs the instrumental variable method to mitigate endogeneity issues arising from reverse causality. Referring to the existing research [89, 90], we select regional communication level as the instrumental variable in this study. This choice is motivated by the influence of digital infrastructure development and communication level in the city where enterprises are located on their digital transformation process. A higher communication level enhances support for information, technology, consumer demand, and other aspects crucial for enterprises, thereby accelerating their digital transformation process. Hence, this variable satisfies the correlation condition of instrumental variables. Additionally, regional communication level primarily reflects micro-level application of information technology and does not directly impact enterprise ESG performance, meeting the exogeneity condition. Specifically, we employ mobile phone penetration rate (per 100 people) in the province where an enterprise operates as a proxy for regional communication level. As shown in Table 5, two-stage regression results using instrumental variables exhibit significantly positive effects consistent with previous findings. The Kleibergen-Paap rk LM statistic is significant at the level of 1%, which passes the underidentification test. The Kleibergen-Paap rk Wald F statistic is 53.288, which is larger than the 16.38 critical value of F test at 10% level in weak instrumental variable identification, and passes the weak instrumental variable test, indicating that the selection of instrumental variables in this paper is reasonable to some extent. To sum up, after considering the endogeneity problem, digital transformation can still promote the improvement of ESG performance.

## Further analysis

**Mechanism test.**   In order to investigate the mechanism underlying company transparency in digital transformation and its impact on corporate ESG performance, we construct models (5) and (6) based on Model (4), following the approach of Wen and Ye (2014) [91], to empirically test the mediating effect as outlined below:

$$Tra_{i,t} = \gamma_0 + \gamma_1 Digital_{i,t} + \sum control_{i,t} + Year + Industry + \varepsilon_{i,t} \tag{5}$$

$$ESG_{i,t} = \xi_0 + \xi_1 Digital_{i,t} + \xi_2 Tra_{i,t} + \sum control_{i,t} + Year + Industry + \varepsilon_{i,t} \tag{6}$$

Columns (4) and (5) of Table 3 show the test results of the action mechanism of digital transformation affecting enterprise ESG performance. Among them, the coefficient before digital transformation in Column (4) is significantly positive, indicating that technological advantages and organizational structure changes brought by digital transformation will significantly improve company transparency. Column (5) is the estimated result of Model 3. The results show that the coefficient of company transparency (Tra) is significantly positive, indicating that company transparency plays an intermediary role in the process of digital transformation affecting the ESG performance of enterprises. The research findings demonstrate that the utilization of digital technology enhances corporate transparency, thereby augmenting the

**Table 5. Tool variable method.**

| VARIABLES | Digital | ESG |
|-----------|---------|-----|
| IV | 0.003*** | |
| | (0.000) | |
| Digital | | 4.848*** |
| | | (1.218) |
| Size | 0.049*** | 2.177*** |
| | (0.013) | (0.129) |
| Lev | -0.354*** | -1.091 |
| | (0.072) | (0.754) |
| Grow | 0.153*** | -1.122*** |
| | (0.035) | (0.347) |
| Cashflow | -0.586*** | 5.938*** |
| | (0.160) | (1.547) |
| Indire | 0.471** | -3.466* |
| | (0.192) | (1.774) |
| Listage | -0.349*** | 2.924*** |
| | (0.039) | (0.576) |
| Top1 | -0.128* | 2.790*** |
| | (0.075) | (0.691) |
| Constant | 2.780*** | -48.852*** |
| | (0.656) | (7.390) |
| Observations | 6,044 | 6,044 |
| R-squared | 0.483 | 0.409 |
| Industry | YES | YES |
| Year | YES | YES |

Note:

\***, ** and * indicate significance at the 1%, 5% and 10% levels.

frequency of interaction between enterprises and external investors as well as improving internal supervision efficiency. Consequently, this engenders both internal and external governance effects, ultimately enhancing corporate ESG performance.

At the same time, Sobel method is used to test the mediating effect, in which the z-value of Sobel test is 3.96, p<0.01, which further verfies H2 hypothesis in this paper.

**Moderating effect test.**   In order to deeply reveal what changes will occur in the effect of Digital transformation on ESG performance when there is performance expectation gap in manufacturing enterprises, this paper constructs the interaction between performance expectation gap and digital transformation (Digital*Ngap), referring to the existing literature [, On the basis of Model (4), construct (7) to test the moderating effect of performance expectation gap:

$$ESG_{i,t} = \mu_0 + \mu_1 Digital_{i,t} + \mu_2 Ngap_{i,t} + \mu_3 Digital_{i,t} \times$$
$$Ngap_{i,t} + \mu_4 \sum control_{i,t} + Year + Industry + \varepsilon_{i,t} \tag{7}$$

The empirical results of the moderating effect are presented in (6) of Table 3, revealing a significantly negative regression coefficient (-2.787) for the interaction term (Digital*Ngap) at a 5% significance level. This indicates that the performance expectation gap does not trigger a "make or break" effect on enterprises. On the contrary, due to the presence of this gap, internal

survival pressures and external investor doubts lead to reduced environmental protection investments and fulfillment of social responsibility by enterprise management.

Furthermore, when actual performance falls short of expectations, disputes may arise within internal management regarding whether to continue with strategic reforms. If there is an entrenched resistance within management circles, managers from different positions face significant threats. In such circumstances, persisting with implementing reform strategies escalates strategic risks and potentially triggers a "stop-loss in time" effect. Additionally, based on limited attention hypothesis, conflicts over internal control rights further divert managerial focus away from utilizing digital transformation's technical advantages to enhance internal governance efficiency–resulting in declining corporate environmental, social, and governance performance levels. The hypothesis H3b in this paper is thus confirmed, indicating that the performance expectation gap plays a negative moderating role in the relationship between digital transformation and corporate ESG realization.

## Heterogeneity analysis

**Nature of property rights.** State-owned enterprises possess inherent advantages in resource acquisition, market competition, innovation strength, strategic reform risk, and other aspects due to their unique institutional advantages [92]. The process of digital transformation requires significant capital investment, the recruitment of digital technology talents, and the implementation of digital technologies. State-owned enterprises enjoy strong credit endorsement which makes financial institutions and external investors prefer supporting them financially [93]. This effectively mitigates the crowding-out effect on innovation behavior, environment, and society caused by dedicated capital investment during enterprise reform.

Moreover, state-owned enterprises' excellent corporate image attracts more talent compared to non-state-owned enterprises, thereby addressing the personnel allocation-technical resources mismatch during digital transformation that hinders leveraging the enabling effect of digital technology. Therefore, as key players in China's ESG system and national strategic policy implementation initiatives, state-owned enterprises are more proactive in improving their ESG performance. By contrast, non-state-owned enterprises prioritize seeking economic benefits through leveraging competitive advantages offered by digital technology amidst fierce market competition and environmental uncertainty. Non-economic benefits are often not their core objective. Therefore, based on this analysis,the promotion effect of digital transformation on ESG performance is significantly greater for state-owned enterprises than for non-state-owned ones.

This study categorizes enterprises into state-owned and non-state-owned based on their ownership nature. Columns (1) and (2) of Table 6 present the regression results for different ownership types. The findings indicate that, in comparison to non-state-owned enterprises, state-owned enterprises exhibit a higher regression coefficient, suggesting a more significant role of digital transformation in enabling state-owned enterprises.

**Intensity of industry competition.** The level of market competition within an industry significantly influences the strategic formulation of enterprises [94]. In highly competitive industries, products exhibit high homogeneity and strong substitutability. When transformative breakthroughs in product innovation are unattainable, enterprises are inclined to leverage digital technology's information resources, organizational changes, business models, and other competitive advantages to enhance non-financial performance in environmental sustainability, social responsibility, and corporate governance. This approach aims to bolster enterprise reputation, cultivate distinctive soft power capabilities, and facilitate differentiation amidst intense market competition [16, 64].

**Table 6. Heterogeneity test.**

| VARIABLES | State-owned enterprises | Non-state-owned enterprises | High competition | Low competition |
|---|---|---|---|---|
| | ESG | ESG | ESG | ESG |
| Digital | 0.305** | 0.246* | 0.419*** | 0.078 |
| | (0.139) | (0.137) | (0.133) | (0.145) |
| Size | 2.478*** | 2.419*** | 2.118*** | 2.675*** |
| | (0.132) | (0.142) | (0.141) | (0.125) |
| Lev | -5.580*** | 0.069 | -3.768*** | -1.975*** |
| | (0.714) | (0.673) | (0.682) | (0.694) |
| Grow | -0.349 | -0.509* | -0.062 | -0.826** |
| | (0.451) | (0.293) | (0.332) | (0.381) |
| Cashflow | 3.888** | 3.962** | 1.086 | 4.249*** |
| | (1.822) | (1.591) | (1.764) | (1.612) |
| Indire | -6.249*** | 2.383 | -4.898** | 1.865 |
| | (2.161) | (2.022) | (1.992) | (2.038) |
| Listage | 0.940* | 0.892** | 1.872*** | 0.597 |
| | (0.570) | (0.354) | (0.363) | (0.467) |
| Top1 | 6.093*** | -0.056 | 3.502*** | 0.884 |
| | (0.887) | (0.828) | (0.828) | (0.831) |
| Constant | -33.667*** | -40.501*** | -25.086*** | -38.910*** |
| | (4.133) | (3.083) | (3.162) | (4.145) |
| Observations | 2,747 | 3,297 | 3,016 | 3,028 |
| R-squared | 0.603 | 0.574 | 0.575 | 0.578 |
| Industry | YES | YES | YES | YES |
| Year | YES | YES | YES | YES |

Note:

***, ** and * indicate significance at the 1%, 5% and 10% levels.

Therefore, this paper argues that the impact of digital transformation on enterprise ESG performance is more pronounced in highly competitive industries. To test this hypothesis, we adopt the established research methodology [95] and employ the Herfindahl index (the sum of squared ratios of each company's main business income to the total main business income of the industry) as a measure of industry competition intensity. The regression results are presented in columns (3) and (4) of Table 6. The findings indicate that low competition does not yield statistically significant results, whereas high competition exhibits a significantly positive effect at a 1% level of significance. This suggests that in highly competitive environments, enterprises can achieve more substantial improvements in their ESG performance through leveraging digital technology.

## Discussion

Firstly, the empirical analysis results confirm the hypothesis (H1) proposed in this study. This finding aligns with existing research and further substantiates that digital transformation not only positively impacts financial performance but also serves as an internal driver for enhancing ESG performance within enterprises [13, 25, 26].

Furthermore, this study confirms the proposition H2. Existing literature predominantly examines the relationship between digital transformation and ESG performance through the lenses of total factor productivity and dynamic capability, neglecting the role of corporate

transparency in this context. On one hand, the application of digital technology brings about technological advantages that generate a "governance effect," enhancing internal governance capabilities by increasing shareholder participation in decision-making and curbing managerial discretion [48]. On the other hand, digital transformation yields a "spotlight effect" that amplifies market attention towards enterprises and facilitates greater interaction frequency of internal and external information [59], thereby promoting environmental and social investments among manufacturing firms to enhance their ESG performance. This research finding expands upon existing knowledge regarding the mechanisms linking digital transformation with enterprise ESG performance while contributing to non-economic value research within the domain of digital transformation.

This study also investigates whether the relationship between digital transformation and ESG performance is influenced by the performance expectation gap, and the empirical findings confirm the hypothesis H3b proposed in this study. The underlying reason is that during a performance expectation gap, enterprises face increased strategic risks and heightened internal and external pressures on management [61, 62]. Pursuing strategic reforms at such times may not yield immediate turnaround results but can potentially lead to organizational difficulties. Consequently, the performance expectation gap tends to foster more conservative strategic decision-making by management, thereby limiting the extent to which digital transformation can promote ESG performance. This conclusion underscores the significance of performance feedback in understanding the intrinsic connection between digital transformation and ESG performance within enterprises while addressing existing research limitations.

## Conclusion

In the face of medium and long-term constraints posed by the 'dual carbon' goal, leveraging competitive advantages brought about by digital transformation to stimulate ESG practice motivation and help enterprises explore a sustainable development path with economic and social benefits has become a major concern for academia and industry. While scholars have begun exploring the impact mechanism and effect of digital transformation on enterprise ESG performance, the 'black box' remains unopened, with impact boundaries yet to be fully revealed. Therefore, this paper empirically investigates the impact of digital transformation on ESG performance in manufacturing industries, elucidating its internal mechanisms from a company transparency perspective while revealing differences in relationships between digital transformation and ESG performance under conditions of performance expectation gaps. This study provides new theoretical references and policy implications for deep integration between digital transformation and green transformations. The findings demonstrate that: (1) Digital transformation has a significant positive impact on enterprise ESG performance.(2) Analysis of the influence mechanism reveals that company transparency partially mediates the relationship between digital transformation and enterprise ESG performance. (3) The performance expectation gap will give rise to the phenomenon of "timely stop loss" and impede the transformative impact of digitalization on the ESG performance of manufacturing enterprises. (4) Through heterogeneity analysis of the internal and external environment, it is observed that in highly competitive industries within the external environment, digital transformation exhibits a more pronounced positive influence on enterprise ESG performance. State-owned enterprises can fully leverage the enabling role of digital transformation.

### Theoretical and practical contributions

The exploration of ESG practice in emerging markets holds significant theoretical significance for the advancement of the ESG field [96]. Despite China's rapid development as an

emerging economy, its research in the realm of ESG is still nascent [97]. By focusing on China as a research subject, this study not only expands the investigation into influencing factors on ESG within China but also offers insights applicable to sustainable development in other developing nations. Moreover, from a corporate transparency perspective, this study elucidates the logical framework linking digital transformation and enterprise ESG performance while broadening our understanding of how digital transformation impacts such performance. It also explores the significance of the performance expectation gap in the internal relationship between digital transformation and ESG performance, thereby addressing the limitations of existing research.

From a practical perspective, this study unveils the mechanism and impact of digital transformation on the ESG performance of enterprises, offering a novel empirical reference for effectively aligning digitization with environmental sustainability efforts in underperforming companies. Moreover, it provides fresh insights for governments to formulate incentivizing policies. Specifically, organizations need to shift their development mindset and fully recognize the long-term advantages of investing in environmental, social, and governance initiatives. Simultaneously, careful attention should be paid to potential adverse effects arising from digital transformation; thus necessitating timely adjustments in personnel allocation, organizational structure, and business processes to ensure optimal utilization of digital technologies' enabling capabilities. Furthermore, The government should prioritize the impact of altruistic preferences [98] and develop a robust policy incentive framework encompassing capital infusion, talent cultivation, and equipment provisioning. This will help alleviate resource scarcity-induced reluctance or apprehension towards ESG investments during the process of enterprise digitization while partially sharing change-related risks.

## Limitation and future research

There are certain limitations in this study. Despite employing text analysis and utilizing data from the CSMAR database to measure the extent of enterprise digital transformation, it is still unable to completely mitigate the influence of management behavior such as false disclosure and exaggeration, which may introduce some deviation between the measurement indicators and the actual scenario. Future research could explore alternative measurement methods to minimize potential errors. Furthermore, due to data constraints, this study does not investigate the effects of the COVID-19 outbreak on enterprises' ESG practices; Therefore, future studies can explore disparities in impact before and after the outbreak. Lastly, it is important to note that our sample only encompasses Chinese market enterprises with ESG rating agency coverage and does not encompass emerging markets comprehensively. Subsequent research could concentrate on discerning differences between digital transformation and ESG performance in developed versus developing countries.

## Supporting information

**S1 Data.**
(XLSX)

## Acknowledgments

We would like to express our heartfelt gratitude to the anonymous reviewers, editor, and everyone who contributed to the writing of this paper.

## Author Contributions

**Conceptualization:** Xianyun Wu, Longji Li.

**Data curation:** Xianyun Wu, Longji Li.

**Formal analysis:** Longji Li.

**Funding acquisition:** Xianyun Wu.

**Investigation:** Longji Li.

**Methodology:** Longji Li.

**Project administration:** Longji Li.

**Software:** Longji Li.

**Supervision:** Xianyun Wu.

**Visualization:** Longji Li.

**Writing – original draft:** Xianyun Wu, Longji Li.

**Writing – review & editing:** Xianyun Wu, Dekuan Liu, Qian Li.

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
