## [Decision Letter · Decision Letter 0]

8 Jan 2024

PONE-D-23-39817Technology empowerment: Digital transformation and enterprise ESG performance--Evidence from China's manufacturing sectorPLOS ONE

Dear Dr. Li,

Thank you for submitting your manuscript to PLOS ONE. After careful consideration, we feel that it has merit but does not fully meet PLOS ONE’s publication criteria as it currently stands. Therefore, we invite you to submit a revised version of the manuscript that addresses the points raised during the review process.

We look forward to receiving your revised manuscript.

Kind regards,

Jianhua Zhu

Academic Editor

PLOS ONE

Journal Requirements:

 This study obtained the research results of Liaoning Economic and Social Development Research Project in 2024 by the Provincial Federation of Social Sciences (Project No.:2024lslybwzzkt-034) and Education project of Liaoning Social Science Planning Fund "Research on Improving the Learning Effect of Cross-school Courses based on fsQCA"(Project No.:L21AED005).

3. In the online submission form, you indicated that The data presented in this study are available on request from the corresponding author.

Reviewers' comments:

Reviewer's Responses to Questions

**Comments to the Author**

1. Is the manuscript technically sound, and do the data support the conclusions?

Reviewer #1: Yes

Reviewer #2: Partly

2. Has the statistical analysis been performed appropriately and rigorously? 

Reviewer #1: Yes

Reviewer #2: No

3. Have the authors made all data underlying the findings in their manuscript fully available?

Reviewer #1: Yes

Reviewer #2: No

4. Is the manuscript presented in an intelligible fashion and written in standard English?

Reviewer #1: Yes

Reviewer #2: Yes

5. Review Comments to the Author

Reviewer #1: This paper studies the effects of digital transformation on ESG from new perspectives company transparency and performance expectation, which is not studied yet in the existing literature. And the paper is well-written.

There are a few concerns as follows.

1. This paper finds company transparency as the major mechanism channel. As the authors put, existing literature has uncovered other channels. Therefore, I suggest the authors to do some work to prove why the channel emphasized by this paper is important.

2. References should be added properly when analyzing the mechanism at Section 2.

3. Why not control firm fixed effects in the baseline model. And the year fixed effects should be included even when controlling firm fixed effects.

4. The economic significance of the core estimated coefficients in the baseline results (Table 3) should be added.

Reviewer #2: Abstract

An abstract typically provides a concise summary of the main points of a research paper, including its purpose, methodology, results, and conclusions.

Remove all the numberings in the manuscript

Introduction

Objectives should be clearly stated as well that highlights all the issues that will be incorporated in this study.

Various studies have been conducted on these concepts, what is new in this study?

Explain the gaps and justify the objectives of the study.

Explain the core issues and the evidence especially statistics and with current literatures to support the issues of this study.

Explain the missing link in the discussions about the way in which countries manage situations related to study but also a gap in academic knowledge.

Identify the research problem as with any academic study, you must state clearly and concisely the research problem that is being investigated.

Appropriate background information has not been provided with special terms and concepts defined.

Objective of this study has not been clearly mentioned?

Lacks research topic or problem not clearly stated shown to be worth investigating as there were many studies conducted, and therefore need to highlight the extension from previous studies.

Structure of the paper is not necessary as this is not a thesis.

What is the novelty of this paper?

Introduction is too long windowed and too many unnecessary explanation, author should summarise it and explain it in a clear manner for reader to understand

Literature Review title is totally missing from the study.

Have a subtitle – Literature review so that the reader able to know what the content in that particular scope

Need to strengthen the literature review.

Places each source in the context of its contribution to the understanding of the specific issue, area of research, or theory under review.

Describes the relationship of each source to the others that you have selected.

Identifies new ways to interpret, and shed light on any gaps in, previous research.

Review scholarship on the topic, synthesizing key themes, and, if necessary, noting studies that have used similar methods of inquiry and analysis.

Note where key gaps exist and how your study helps to fill these gaps or clarifies existing knowledge.

The author should have discussed the issues in detail and how the issues were not resolved or partially resolved by previous studies.

Discuss the important recent extensions that have been made to the model to make it more realistic and give a brief overview of some of the older and more recent empirical studies that have fitted the model. Such contributions add up the value of the paper.

Any relationship between the scope and theories also needs to be discussed.

Literature review is lacking the in-depth of the study.

Have a table in appendix to explain the jargon used.

There were so many variables used in the analysis but most of them we not rigorously discussed in literature review

Has this study undertaken the consequences of pre and post covid situation in terms of vulnerability in country’s economy?

Methodology

Explanation of mediating, moderating independent and dependent variables should be explained in literature review and not in research methodology

Methodology is lacking in the study.

A-share listed manufacturing companies in 276 Shanghai and Shenzhen – justify why only 2 cities used in this study

6044 observation – justify why such a big sample required and any measurement was conducted

Collection of data for the time period need to reflect the situation of post pandemic.

What are the sources of data

Explanation of methodology is weak.

Lack of evidence of care and accuracy in the data process

Unable to reveal the research methods fully described of the advantages and disadvantages of chosen methods that was discussed.

What were the instruments used in the study

The chosen data need to be justified from the sources and why other sources are not utilised.

Credibility evaluation must be performed to understand the authenticity of the information available.

Unable to reveal the research methods fully described of the advantages and disadvantages of chosen methods that was discussed.

What software has been used to analyse the data as to look into analysis stetting need to know the software

Analysis

Unable to comment on the analysis path as it depend on what software has been utilised to analyse as different software has different method of analysis

The organization and discussion could be improved quite a bit, to make it clearer in some places to demonstrate symbolic role.

Ensure all analysis undertaken should have evidence of software format in the appendix

Discussion

Discussion has not been incorporated in the manuscript

The finding of the research needs to be compared and contrasted with findings, theories, models and concepts derived from the literature review.

The relevance of the conclusions for stakeholders has not been discussed thoroughly.

Most striking aspect that the study protocol brought forward and among the plethora of texts proposed

Comment on whether or not the results were expected for each set of findings; go into greater depth to explain findings that were unexpected or especially profound. If appropriate, note any unusual or unanticipated patterns or trends that emerged from your results and explain their meaning in relation to the research problem.

Either compare your results with the findings from other studies or use the studies to support a claim. This can include re-visiting key sources already cited in your literature review section.

Describe the patterns, principles, and relationships shown by each major findings and place them in proper perspective. The sequence of this information is important; first state the answer, then the relevant results, then cite the work of others.

Good discussion section includes analysis of any unexpected findings. This part of the discussion should begin with a description of the unanticipated finding, followed by a brief interpretation as to why you believe it appeared and, if necessary, its possible significance in relation to the overall study.

The discussion section should end with a concise summary of the principal implications of the findings regardless of their significance. Give a brief explanation about why you believe the findings and conclusions of your study are important and how they support broader knowledge or understanding of the research problem.

Conclusion

The overall argument has not been summarized.

The reflection on the aims, methods, and results of the research is lacking.

Relevant recommendations have not been discussed.

What are your recommendations to the overall stakeholders?

Conclusions and recommendations discussed in the context need to be widely applicable.

Limitations and future study need to be incorporated as well

Lack of current academic citations

Reference

Too many citations missing for the points stated in the content

Need more current academic citations to reflect the current academic study.

99% of the references are from China which does not represent the global aspect of the literature

6. PLOS authors have the option to publish the peer review history of their article (what does this mean?). If published, this will include your full peer review and any attached files.

Reviewer #1: No

Reviewer #2: **Yes: **Sanmugam Annamalah

---

## [Author Response · Author response to Decision Letter 0]

21 Feb 2024

Dear Editor and reviewers,

We sincerely appreciate your correspondence and the reviewers' insightful comments on our manuscript titled "Technology Empowerment: Digital Transformation and Enterprise ESG Performance - Evidence from China's Manufacturing Sector" (Manuscript ID: PONE-D-23-39817). These valuable comments have significantly contributed to the revision and enhancement of our paper. We have thoroughly examined all the feedback and diligently made revisions accordingly, which are indicated in blue throughout the revised version.

 The major corrections made in response to the reviewers' comments are summarized as follows:

Review Comments to the Author

Reviewer #1:

1.This paper finds company transparency as the major mechanism channel. As the authors put, existing literature has uncovered other channels. Therefore, I suggest the authors to do some work to prove why the channel emphasized by this paper is important.

Reply:Thank you very much indeed for your comments.According to your suggestions, we have incorporated a literature review section into the paper.To underscore the theoretical significance of this study, a greater number of internationally acclaimed scholarly works have been incorporated to substantiate the arguments posited herein.

2.References should be added properly when analyzing the mechanism at Section 2.

Reply:Thank you very much indeed for your comments.According to your suggestions, we have revised the theoretical analysis section of this study by incorporating additional scholarly references to substantiate our research.

3.Why not control firm fixed effects in the baseline model. And the year fixed effects should be included even when controlling firm fixed effects.

Reply:Thank you very much indeed for your comments.To establish the benchmark model and validate H1 proposed in this paper, we adopt a similar approach as previous studies [46,47] by controlling for year and industry fixed effects to enhance the robustness of our model. Additionally, to ensure stable regression results, we follow the methodology employed in prior research [48,49] and employ a multidimensional fixed effect model that includes controls for industry-firm-year fixed effects. Remarkably, even with these adjustments, our regression results consistently support H1 posited in this study.

4.The economic significance of the core estimated coefficients in the baseline results (Table 3) should be added.

Reply:Thank you very much indeed for your comments.According to your suggestions, we have meticulously revised the manuscript and incorporated the economic significance of the core estimation coefficient into the paper.

Reviewer #2:

1.An abstract typically provides a concise summary of the main points of a research paper, including its purpose, methodology , results , and conclusions.

Reply:Thank you very much indeed for your comments.As you mentioned ,We have made the change. The new abstract as follows : In light of the long-term constraints posed by the "dual carbon" objective, can digital technology emerge as a transformative solution for enterprises to embark on a sustainable development trajectory? The existing body of research has yet to reach a consensus. In order to shed further light on the intricate relationship between digital transformation and ESG performance of enterprises, this study empirically examines the mechanisms and boundaries through which digital transformation influences ESG performance, based on observational data from A-share manufacturing listed companies in Shanghai and Shenzhen spanning from 2011 to 2021. The findings demonstrate that digital transformation exerts a significant positive impact on the ESG performance of manufacturing enterprises. Mechanism analysis reveals that the enabling effect of digital transformation primarily enhances corporate transparency, thereby fostering continuous improvements in ESG performance among manufacturing enterprises. The performance expectation gap will give rise to the phenomenon of "stop-loss in time" and impede the promotional impact of digital transformation. Further investigation into industrial characteristics and industry competition intensity indicates that state-owned enterprises and those operating within highly competitive environments experience more pronounced effects of digital transformation on their ESG performance. This study expands the mechanism and boundary of digital transformation on ESG performance of manufacturing enterprises, and provides a new perspective for manufacturing enterprises to realize the collaborative transformation of digital and green.

2.Remove all the numberings in the manuscript.

Reply:Thank you very much indeed for your comments. In our revised manuscript, we have eliminated the paragraph numbering .

3.Objectives should be clearly stated as well that highlights all the issues that will be incorporated in this study.

Reply:Thank you very much indeed for your comments.According to your suggestion, we have revised the study objectives.The details are as follows:By examining the existing research findings in the domains of digital transformation and ESG performance, it is evident that while the relationship between digital transformation and ESG performance has been established by current research, further exploration is required to understand its influencing mechanism and contextual conditions. To comprehensively analyze the impact and prerequisites of digital transformation on enterprise ESG performance, this study integrates digital transformation, company transparency, and enterprise ESG performance within a unified research framework. It elucidates their internal logic, introduces the concept of a performance expectation gap, and investigates whether the mechanism of digital transformation alters under circumstances of declining performance to offer novel evidence for addressing these aforementioned issues.

4.Various studies have been conducted on these concepts, what is new in this study?

Reply:Thank you very much indeed for your comments.The present study makes the following contributions:By reviewing the existing research findings in the domains of digital transformation and enterprise ESG performance, this study reveals that while current literature has established a confirmed association between digital transformation and enterprise ESG performance, further exploration is warranted to elucidate its underlying mechanisms and boundaries. To comprehensively investigate the impact and limitations of digital transformation on manufacturing enterprise ESG performance, this paper adopts company transparency as a focal point for empirical analysis to unravel the influencing mechanism of digital transformation. Additionally, incorporating the concept of performance expectation gap into our research framework enables us to explore whether its existence affects the efficacy of digital transformation.This study further reveals the mechanism and boundary of digital transformation , while employing multiple data sources to mitigate potential deviations arising from false information disclosure.This offers a novel perspective for manufacturing enterprises seeking to achieve collaborative digitalization and greening initiatives.

5.Explain the gaps and justify the objectives of the study.

Reply:Thank you very much indeed for your comments.By synthesizing research findings in related fields, it is evident that there are still certain limitations in the existing literature: (1) the existing research mainly discusses the influence between the two from the perspective of internal control, green innovation and information disclosure quality, and its internal influence mechanism needs to be further expanded. (2) The measurement approach for assessing the extent of digital transformation within enterprises remains singular, making it challenging to mitigate potential deviations resulting from false corporate disclosures. (3) What are the requisite conditions for effectively harnessing the impact of digital transformation empowerment?

To address the aforementioned challenges and further elucidate the mechanisms and boundaries of digital transformation on corporate ESG performance, this study adopts company transparency as a focal point and incorporates the performance expectation gap into our research framework to investigate the Realistic question of whether digital transformation can incentivize enterprises to engage in ESG performance.

6.Explain the core issues and the evidence especially statistics and with current literatures to support the issues of this study.

Reply:Thank you very much indeed for your comments.In response to the global green transformation of the economy, China proposed a "dual carbon" target in 2020, which sets the direction for China's economic development in the coming decade. The manufacturing industry, as a key pillar of national economic growth, is also home to enterprises with high energy consumption, pollution levels, and emissions [1]. Promoting the green transformation of this industry is crucial for achieving the "double carbon" goal. However, amidst limited resources, talent pool, and funding availability, striking a balance between economic and social benefits has become a focal point of current academic discourse. This research addresses this core issue by exploring how digital technology can contribute to enterprise ESG performance .

7.Explain the missing link in the discussions about the way in which countries manage situations related to study but also a gap in academic knowledge.

Reply:Thank you very much indeed for your comments，This is indeed a great help to our paper.We have incorporated pertinent material into the revised paper，The specific correction is as follows:Currently, the global wave of low-carbon transformation is underway, with all countries worldwide implementing ESG-related policies and regulations. Examples include the Corporate Sustainability Reporting Directive and IFRS S1 - General Requirements for Disclosure of Information Sustainability-related Financial Information. In comparison to European and American nations, China's ESG policy system is still in its nascent stage, with only mandatory disclosure regulations on ESG information being issued for Chinese central State-owned enterprises. Within the academic realm, China's research on ESG started relatively late compared to the more matured studies on ESG investment and practice in European and American countries [2]. Scholars in China primarily focus on examining the impact of ESG information disclosure as well as establishing an ESG rating system[3].

8.Identify the research problem as with any academic study, you must state clearly and concisely the research problem that is being investigated.

Reply:Thank you very much indeed for your comments.After reviewing the existing literature in the field of ESG, it becomes evident that current research predominantly focuses on the economic value derived from ESG and external factors influencing ESG performance, while neglecting to address how to internally stimulate the endogenous driving force for enhancing ESG performance within enterprises.

The application of information technology offers a novel perspective for addressing this issue. Previous studies have provided initial evidence on the relationship between digital transformation and enterprise ESG performance. However, given the current global economic turmoil, further investigation is required to understand the mechanisms underlying digital transformation, as well as whether its impact changes in light of the performance expectation gap. Therefore, this study focuses on China's emerging economy by selecting observational data from A-share manufacturing listed companies in Shenzhen and Shanghai stock exchanges. Specifically, it aims to examine the mechanism of digital transformation on the ESG performance of manufacturing enterprises, thereby providing new empirical evidence to address these aforementioned concerns.

9.Appropriate background information has not been provided with special terms and concepts defined.

Reply:Thank you very much indeed for your comments.According to your suggestions, we have made the following modifications to the background section.

In 2004, the United Nations introduced the concept of ESG (Environmental, Social, and Governance) in its initiative report titled "Who Cares Wins" .This report provided a new direction for businesses on how to implement sustainable development principles. The concept of ESG originates from ethical investment and responsible investment, rejecting the profit-centric business philosophy, and advocating for enterprises to incorporate environmental, social, and governance factors into their investment decisions while considering economic benefits[4-6]. Currently, there is a global wave of low-carbon transformation underway, leading all countries worldwide to introduce ESG-related policies and regulations. Examples include the Corporate Sustainability Reporting Directive and IFRS S1 - General Requirements for Disclosure of Information Sustainability-related Financial Information. In recent years, China's "dual carbon" goal has accelerated the development process of ESG in China [7]. Regulators have issued a series of policies and regulations that gradually require listed companies to disclose ESG-related information; thus making ESG practices an essential aspect for enterprise development. However, challenges such as insufficient willingness and limited participation in specific corporate practices undermine the positive impact of the ESG system on China's economic transformation. Therefore, it is crucial to explore both internal and external factors influencing enterprises' performance in implementing ESG.

10.Objective of this study has not been clearly mentioned ? Lacks research topic or problem not clearly stated shown to be worth investigating as there were many studies.

Reply:Thank you very much indeed for your comments.The primary objective of this study is to investigate the mechanism and conditions that drive digital transformation in enhancing enterprise ESG performance. In response to your suggestion, we have revised the introduction section of this study.

Thank you again for your positive comments and valuable suggestions to improve the quality of our manuscript.

11.Structure of the paper is not necessary as this is not a thesis

Reply:Thank you very much indeed for your comments.According to your suggestion, we have omitted this section of the content.

12.What is the novelty of this paper?

Reply:Compared to previous studies, this study innovatively addresses the following aspects: (1) Previous studies did not investigate whether the relationship between digital transformation and ESG performance of enterprises would be influenced during periods of declining enterprise performance. By introducing the situational condition of performance period gap, this study further defines the impact of digital transformation on ESG performance and enriches research on between digital transformation and performance feedback. (2) From a corporate transparency perspective, this paper elucidates the mechanism through which digital transformation affects ESG performance in manufacturing enterprises, offering new theoretical references and practical insights for sustainable development enabled by digital technology.

13.Introduction is too long windowed and too many unnecessary explanation, author should summarise it and explain it in a clear manner for reader to understand

Reply:Thank you very much indeed for your comments.According to the issues raised in the introductory section of the article, we have implemented comprehensive modifications. The specific revisions are outlined as follows:

In 2004, the United Nations introduced the concept of ESG (Environmental, Social, and Governance) in its initiative report titled "Who Cares Wins" .This report provided a new direction for businesses on how to implement sustainable development principles. The concept of ESG originates from ethical investment and responsible investment, rejecting the profit-centric business philosophy, and advocating for enterprises to incorporate environmental, social, and governance factors into their investment decisions while considering economic benefits[4-6]. Currently, there is a global wave of low-carbon transformation underway, leading all countries worldwide to introduce ESG-related policies and regulations. Examples include the Corporate Sustainability Reporting Directive and IFRS S1 - General Requirements for Disclosure of Information Sustainability-related Financial Information. In recent years, China's "dual carbon" goal has accelerated the development process of ESG in China [7]. Regulators have issued a series of policies and regulations that gradually require listed companies to disclose ESG-related information; thus making ESG practices an essential aspect for enterprise development. However, challenges such as insufficient willingness and limited participation in specific corporate practices undermine the positive impact of the ESG system on China's economic transformation. Therefore, it is crucial to explore both internal and external factors influencing enterprises' performance in implementing ESG.

At present, China is in a critical period of transformation from a manufacturing power to a manufacturing power. Manufacturing is the backbone of the country's economic development , Facing the medium and long term constraints of "dual carbon" target, whether manufacturing enterprises can explore a sustainable transformation path is related to the long-term healthy development of China's economy[8]. The wave of digital transformation offers a novel perspective for the sustainable development of manufacturing enterprises. Digital transformation is regarded as the extensive application of digital technology across various aspects of enterprise survival, operation, and sales [9]. Previous studies have demonstrated that the adoption of digital technology can enhance the economic efficiency of manufacturing enterprises by improving resource allocation efficiency, innovation capability, and customer information advantage [10-12] . However, can the technological advancements and resource utilization resulting from digital transformation effectively stimulate the inherent capabilities of manufacturing enterprises to enhance their environmental, social, and governance (ESG) performance? Although previous studies have made preliminary explorations into the relationship between digital transformation and ESG performance [13，14], the mechanism underlying digital transformation remains incompletely elucidated, necessitating further exploration of working conditions. Therefore, this study aims to further expand the existing research on this topic in order to address the limitations identified in previous studies.

Building upon China's "dual carbon" goal policy context, this study delves into the potential of digital transformation in the manufacturing industry to stimulate endogenous drivers for enhancing ESG performance within enterprises. This investigation aims to unveil the underlying mechanisms of digital transformation, augment existing research findings, and hold significant theoretical and practical implications. Consequently, this study adopts corporate transparency as a foundational aspect and integrates the performance expectation gap into its research framework. Empirical analysis is conducted using observation data from A-share manufacturing listed companies on Shanghai and Shenzhen Stock Exchanges spanning from 2011 to 2021 to examine the boundaries and mechanisms through which digital transformation influences corporate ESG performance.Compared to previous studies, this study innovatively addresses the following aspects: (1) Previous studies did not investigate whether the relationship between digital transformation and ESG performance of enterprises would be influenced during periods of declining enterprise performance. By introducing the situational condition of performance period gap, this study further defines the impact of digital transformation on ESG performance and enriches research on between digital transformation and performance feedback. (2) From a corporate transparency perspective, this paper elucidates the mechanism through which digital transformation affects ESG performance in manufacturing enterprises, offering new theoretical references and practical insights for sustainable development enabled by digital technology.

14.Literature Review title is totally missing from the study.

Have a subtitle – Literature review so that the reader able to know what the content in that particular scope.

Need to strengthen the literature review.

Places each source in the context of its contribution to the understanding of the specific issue, area of research, or theory under review.

Describes the relationship of each source to the others that you have selected.

Identifies new ways to interpret, and shed light on any gaps in, previous research.

Review scholarship on the topic, synthesizing key themes, and, if necessary, noting studies that have used similar methods of inquiry and analysis.

Note where key gaps exist and how your study helps to fill these gaps or clarifies existing knowledge.

The author should have discussed the issues in detail and how the issues were not resolved or partially resolved by previous studies.

Discuss the important recent extensions that have been made to the model to make it more realistic and give a brief overview of some of the older and more recent empirical studies that have fitted the model. Such contributions add up the value of the paper.

Any relationship between the scope and theories also needs to be discussed.

Literature review is lacking the in-depth of the study.

Have a table in appendix to explain the jargon used.

There were so many variables used in the analysis but most of them we not rigorously discussed in literature review.

Reply:Thank you very much indeed for your comments.According to your suggestion, we have incorporated a comprehensive literature review section into the paper. The specific modifications are outlined below.

Since the inception of the ESG concept in 2004, it has garnered significant attention from investors and business managers owing to its unique ability to balance economic benefits with social values. Consequently, academic research in ESG-related fields has witnessed substantial growth[15], with scholars predominantly favoring investigations into the impact of ESG[16]. Mainstream scholars contend that ESG practices can enhance enterprise brand valuation and foster green innovation capabilities, thereby mitigating business risks and ultimately improving enterprise value [17-20]. Scholars have also started examining the influencing factors of enterprise ESG performance. Previous research indicates that factors such as regional digital finance development and environmental protection tax legislation can significantly contribute to enhancing enterprise ESG performance [21,22]. However, existing studies pay more attention to the external factors that affect the ESG performance of enterprises. In order to fully play the positive role of ESG system in the low-carbon transformation of Chinese enterprises, it is necessary to stimulate the endogenous motivation of enterprises to improve ESG performance.

 With the advent of a new wave of scientific and technological revolution, digital technologies such as big data and blockchain offer a novel avenue for facilitating the high-quality development of manufacturing enterprises. Esteemed scholars contend that leveraging digital technology can enhance resource allocation efficiency, innovation capabilities, and customer information advantage, thereby fostering the high-quality development of manufacturing enterprises [10,23,24].In addition to researching the economic benefits of digital transformation, scholars have also begun to focus on its non-economic value. Specifically, they argue that the application of digital technology can facilitate green innovation in enterprises and lead to a reduction in carbon emissions [25,26].With the advancement of research, scholars have started to establish a connection between digital transformation and enterprise ESG performance, leading to two main categories in existing research findings：the "empowerment" effect and the “too much is not good” effect . The "empowerment" effect is specifically reflected in the fact that digital transformation can improve the ESG performance of enterprises by reducing agency costs and improving corporate reputation and dynamic capabilities [27,28]. The “too much is not good” effect is specifically reflected in the fact that a high level of digitalization may weaken the ability and motivation of enterprises to carry out ESG practices. Asymmetric digital transformation and organizational transformation process make it difficult to play the enabling effect of digital technology, which may lead to "information overload" and reduce the information processing ability of enterprises. In addition, a large amount of capital investment in the materialization of digital technology may induce "crowding-out effect" and delay the process of enterprise green transformation [29-31]. 

The concept of transparency emerged from research in the field of information disclosure [32]. As research on information disclosure expanded, scholars introduced the notion of "company transparency," which refers to providing specific company information to external stakeholders [33].With the deepening of research, Chinese scholars have refined the concept of company transparency, that is, the higher the transparency of a company, the wider and deeper the scope and level of external investors' access to internal information of a company, and the stronger the liquidity of information [34].The application of digital technology offers a novel perspective for researching company transparency. However, upon reviewing existing literature, it is evident that scholars tend to associate digital transformation with analysts' forecasts and corporate governance [35,36]. These studies suggest that while there may be a close relationship between digital transformation and company transparency, further exploration is necessary.

The aforementioned analysis reveals that despite the existence of relevant studies demonstrating the correlation between digital transformation and ESG performance, certain limitations persist, primarily in the following aspects: (1) the existing research mainly discusses the influence between the two from the perspective of internal control, green innovation and information disclosure quality, and its internal influence mechanism needs to be further expanded. (2) The measurement approach for assessing the extent of digital transformation within enterprises remains singular, making it challenging to mitigate potential deviations resulting from false corporate disclosures. (3) What are the requisite conditions for effectively harnessing the impact of digital transformation empowerment?

Building upon this premise, the present study adopts corporate transparency as a focal point, integrates the performance expectation gap into the research framework, and explores whether digital transformation can incentivize enterprises to engage in ESG practices. The present study contributes to the existing literature on the mechanisms of digital transformation, elucidates the impact of digital transformation in situations characterized by performance expectation gaps, and addresses a research gap in this domain.

15.Has this study undertaken the consequences of pre and post covid situation in terms of vulnerability in country’s economy?

Reply:Thank you very much indeed for your comments.We acknowledge the potential impact of the COVID-19 outbreak on national economic development, as well as its significance for the digital transformation of enterprise ESG performance. However, a comprehensive analysis of this aspect falls beyond the scope of our paper. The objective of this study is to explore the internal relationship between digital transformation and ESG performance in enterprises, elucidate the underlying mechanisms connecting digital transformation, corporate transparency, and ESG performance, and unveil how these effects may vary under conditions of a performance expectation gap. While we recognize that addressing the impact of COVID-19 would have been relevant in our research, it should be noted that our sample period spans from 2011 to 2021. The official announcement declaring an end to the COVID-19 epidemic in China was made on January 8th, 2023 through 'The Overall Plan for Implementing Class B and B Tube Measures for Novel Coronavirus Infection' issued by the Chinese government. Considering that economic impacts resulting from external events are typically reflected with some lag in data availability, we were unable to obtain up-to-date annual reports for enterprises in 2023 at present. This limitation is acknowledged within our discussion on shortcomings and future prospects; thus, we hope subsequent studies can address this deficiency.

16.Explanation of mediating, moderating independent and dependent variables should be explained in literature review and not in research methodology.

Reply:Thank you very much indeed for your comments.According to your suggestions, we have incorporated the core variables of this study into the literature review section, providing a comprehensive overview. In the research methods section, we have meticulously outlined the measurement methodology for these core variables by drawing upon existing scholarly works.

17.Methodology is lacking in the study.

Reply:Thank you very much indeed for your comments.Revised based on your suggestions, we have made adjustments to the methods and model settings employed in Section 4 Research Design. Additionally, we have incorporated a greater number of relevant literature sources to bolster our research design.

18.A-share listed manufacturing companies in 276 Shanghai and Shenzhen – justify why only 2 cities used in this study.

Reply:Thank you very much indeed for your comments.The country chosen for this study is China, which serves as a rapidly emerging economy. Analyzing its corporate ESG practices holds significant reference value for developing countries worldwide. The research samples selected in this study are manufacturing enterprises listed in Shanghai Stock Exchange and Shenzhen Stock Exchange, not only enterprises in Shanghai and Shenzhen. We apologize for any ambiguity in our previous statement and have made modifications based on your suggestions.

19.6044 observation – justify why such a big sample required and any measurement was conducted.

Reply:Thank you very much indeed for your comments.The sample size used in this study is primarily determined based on the number of manufacturing enterprises listed in China's A-shares and the selected observation period, i.e., 2011-2021. To ensure consistency with previous studies[37], we followed a screening process to obtain an initial sample of 6044 as follows: (1) Excluding samples with ST and *ST designations in the current year; (2) Eliminating samples with missing core variable data; (3) Excluding samples with less than three consecutive years of available data.

20.What are the sources of data.

Reply:Thank you very much indeed for your comments.The original financial data utilized in this study, as well as the robustness test concerning the extent of digital transformation, were sourced exclusively from the China Stock Market & Accounting Research Database(CSMAR). Furthermore, the word frequency analysis pertaining to digital transformation primarily relied upon annual reports disclosed by listed companies through Juchao Consulting Network, Shenzhen Stock Exchange, and Shanghai Stock Exchange.

21.Explanation of methodology is weak.

Lack of evidence of care and accuracy in the data process.

Unable to reveal the research methods fully described of the advantages and disadvantages of chosen methods that was discussed.

Reply:Thank you very much indeed for your comments.The measurement method employed in this study for assessing the extent of digital transformation primarily draws upon existing research[38], utilizing the construction of digital dictionaries and text analysis to determine the degree of digital transformation within enterprises. In contrast to previous approaches that relied on intangible assets related to digital technology, questionnaire surveys, and ERP system applications[39-41], this measurement method establishes a relatively objective and comprehensive digital term dictionary based on semantic expressions found in national policies pertaining to the digital economy. Subsequently, it employs text analysis techniques to construct a more holistic indicator reflecting the level of digitization among Chinese enterprises.

The selection of ESG variables mainly adopts Bloomberg ESG rating data. Bloomberg ESG data and solutions are at the forefront of the market, fully combining the specific market conditions in China, and restoring the ESG performance of Chinese manufacturing enterprises to the greatest extent.

The measurement methodology for the Company transparency index is developed by integrating existing research content and aligning it with the research topic[42,43], thereby selecting multiple indicators to comprehensively measure and accurately reflect the true extent of Company transparency.

The measurement method of performance expectation gap primarily encompasses the practices employed by both Chinese and foreign scholars[44,45], wherein the disparity between actual enterprise performance and anticipated performance is utilized as a metric. Numerous scholars have also conducted empirical examinations to validate these specific measurement methods.

22.What were the instruments used in the study.

What software has been used to analyse the data as to look into analysis stetting need to know the software

Reply:Thank you very much indeed for your comments.The study was conducted without the use of any specialized instruments, and data analysis was performed using Python and Stata version 16.0 software.

23.The chosen data need to be justified from the sources and why other sources are not utilised.

Credibility evaluation must be performed to understand the authenticity of the information available.

Reply:Thank you very much indeed for your comments.The original financial data utilized in this study, as well as the robustness test concerning the extent of digital transformation, were sourced exclusively from the China Stock Market & Accounting Research Database(CSMAR).The CSMAR database, developed by Shenzhen Guotai 'an Education Technology Co., Ltd., is widely recognized as the largest, most accurate, and comprehensive economic and financial research database in China. It has been designed to meet the needs of academic research and adheres to professional standards comparable to renowned international databases such as CRSP and Standard & Poor's Compustat. Chinese scholars unanimously acknowledge the authenticity and reliability of the data contained in this database, which effectively reflects enterprise information with utmost accuracy.

24.Unable to comment on the analysis path as it depend on what software has been utilised to analyse as different software has different method of analysis.

Reply:Thank you very much indeed for your comments.The data analysis software utilized in this study is version 16.0 of Stata.

25.The organization and discussion could be improved quite a bit, to make it clearer in some places to demonstrate symbolic role.

Ensure all analysis undertaken should have evidence of software format in the appendix

Reply:Thank you very much indeed for your comments.According to your suggestions, we have thoroughly revised the empirical analysis of the fifth section and have submitted the original data utilized in this study as a separate document to ensure the veracity of our research findings.

26.Discussion has not been incorporated in the manuscript

The finding of the research needs to be compared and contrasted with findings, theories, models and concepts derived from the literature review.

The relevance of the conclusions for stakeholders has not been discussed thoroughly.

Most striking aspect that the study protocol brought forward and among the plethora of texts proposed

Comment on whether or not the results were expected for each set of findings; go into greater depth to explain findings that were unexpected or especially profound. If appropriate, note any unusual or unanticipated patterns or trends that emerged from your results and explain their meaning in relation to the research problem.

Either compare your results with the findings from other studies or use the studies to support a claim. This can include re-visiting key sources already cited in your literature review section.

Describe the patterns, principles, and relationships shown by each major findings and place them in proper perspective. The sequence of this information is important; first state the answer, then the relevant results, then cite the work of others.

Good discussion section includes analysis of any unexpected findings. This part of the discussion should begin with a description of the unanticipated finding, followed by a brief interpretation as to why you believe it appeared and, if necessary, its possible significance in relation to the overall study.

The discussion section should end with a concise summary of the principal implications of the findings regardless of their significance. Give a brief explanation about why you believe the findings and conclusions of your study are important and how they support broader knowledge or understanding of the research problem.

Reply:Thank you very much indeed for your comments.According to your suggestion, we have made detailed modifications to the conclusion section of the paper and incorporated a discussion segment. Drawing upon the description of the study's findings, we conducted a comparative analysis with existing research and elucidated on both theoretical and practical implications, thereby offering novel empirical references for various stakeholders such as government bodies and enterprises. The specific modifications are indicated in blue within the " discussion and conclusion" section of the revised draft.

27.The overall argument has not been summarized.

The reflection on the aims, methods, and results of the research is lacking.

Relevant recommendations have not been discussed.

What are your recommendations to the overall stakeholders?

Conclusions and recommendations discussed in the context need to be widely applicable.

Limitations and future study need to be incorporated as well

Lack of current academic citations

Reply:Thank you very much indeed for your comments.According to your suggestions, we have incorporated specific proposals into this study and comprehensively summarized the arguments to ensure accurate representation of the study's content. Additionally, we have extensively referenced existing studies to bolster our arguments. The specific modifications are indicated in blue within the " discussion and conclusion" section of the revised draft.

28.Too many citations missing for the points stated in the content

Need more current academic citations to reflect the current academic study.

99% of the references are from China which does not represent the global aspect of the literature

Reply:Thank you very much indeed for your comments.In accordance with your suggestion, we have restructured the reference section of the paper and incorporated a greater number of international literature sources to bolster our argument.

According to the editor and reviewers’ comments, we have made extensive modifications to our manuscript and supplemented extra data to make our results convincing. Thank you again for your positive comments and valuable suggestions to improve the quality of our manuscript.If there are any other modifications we could make, we would like very much to modify them and we really appreciate your help.

Sincerely,

Longji Li

References

1.Gu Qingkang, Lin Lefen. Efficiency, influencing factors and peak path of carbon emission reduction in manufacturing industry under the "dual carbon" target: Based on panel data analysis of large manufacturing provinces [J].Economic Issues, 2024, (02): 57-63.

2.Halbritter G, Dorfleitner G. The wages of social responsibility—where are they? A critical review of ESG investing[J]. Review of Financial Economics, 2015, 26: 25-35.

3.Hu Jie, Yu Xianrong, Han Yiming. Can ESG rating promote the green transformation of enterprises? -- Verification based on multi-time point differential method [J]. Journal of Quantitative and Technical Economics, 2023, 40 (07): 90-111.

4.Tarmuji I, Maelah R, Tarmuji N H. The impact of environmental, social and governance practices (ESG) on economic performance: Evidence from ESG score[J]. International Journal of Trade, Economics and Finance, 2016, 7(3): 67.

5.Liu Fangyuan, Wu Yunlong. Digital transformation and corporate ESG responsibility Performance under the "dual-carbon" goal: Impact, effect and mechanism [J/OL]. Science and Technology Progress and Countermeasures, 1-10[2024-02-13].

6.Zhang X, Zhang J, Feng Y. Can companies get more government subsidies through improving their ESG performance? Empirical evidence from China[J]. Plos one, 2023, 18(10): e0292355.

7.Xu Fengmin, Jing Kui, Li Xuepeng. Portfolio Research based on ESG integration under the background of "dual carbon" target [J]. Journal of Finance Research, 2023, (08): 149-169.

8.Haraguchi N, Cheng C F C, Smeets E. The importance of manufacturing in economic development: has this changed?[J]. World Development, 2017, 93: 293-315.

9.Wen H, Zhong Q, Lee C C. Digitalization, competition strategy and corporate innovation: Evidence from Chinese manufacturing listed companies[J]. International Review of Financial Analysis, 2022, 82: 102166.

10.Peng Y, Tao C. Can digital transformation promote enterprise performance?—From the perspective of public policy and innovation[J]. Journal of Innovation & Knowledge, 2022, 7(3): 100198.

11.Niu Y, Wen W, Wang S, et al. Breaking barriers to innovation: The power of digital transformation[J]. Finance Research Letters, 2023, 51: 103457.

12.Liang Xiaotian, Wen Zongyu.Digital transformation of manufacturing industry, Customer information advantage and high-quality development [J]. Statistics and Decision Making,2023,39(07):179-183.

13.Wang Yunchen, Yang Ruoyi, He Kang et al. Can Digital transformation improve ESG performance? -- Research based on legitimacy theory and Information asymmetry theory [J]. Securities Market Review,2023(07):14-25.

14.Zhao X, Cai L. Digital transformation and corporate ESG: Evidence from China[J]. Finance Research Letters, 2023, 58: 104310.

15.Gillan S L, Koch A, Starks L T. Firms and social responsibility: A review of ESG and CSR research in corporate finance[J]. Journal of Corporate Finance, 2021, 66: 101889.

16.Wong W C, Batten J A, Mohamed-Arshad S B, et al. Does ESG certification add firm value?[J]. Finance Research Letters, 2021, 39: 101593.

17.Lee M T, Raschke R L, Krishen A S. Signaling green! firm ESG signals in an interconnected environment that promote brand valuation[J]. Journal of Business Research, 2022, 138: 1-11.

18.Fu Q, Zhao X, Chang C P. Does ESG performance bring to enterprises’ green innovation? Yes, evidence from 118 countries[J]. Oeconomia Copernicana, 2023, 14(3): 795-832.

19.Sassen R, Hinze A K, Hardeck I. Impact of ESG factors on firm risk in Europe[J]. Journal of business economics, 2016, 86: 867-904.

20.Fatemi A, Glaum M, Kaiser S. ESG performance and firm value: The moderating role of disclosure[J]. Global finance journal, 2018, 38: 45-64.

21.Mu W, Liu K, Tao Y, et al. Digital finance and corporate ESG[J]. Finance Research Letters, 2023, 51: 103426.

22.He Y, Zhao X, Zheng H. How does the environmental protection tax law affect firm ESG? Evidence from the Chinese stock markets[J]. Energy Economics, 2023, 127: 107067.

23.Niu Y, Wen W, Wang S, et al. Breaking barriers to innovation: The power of digital transformation[J]. Finance Research Letters, 2023, 51: 103457.

24.Liang Xiaotian, Wen Zongyu.Digital transformation of manufacturing industry, Customer information advantage and high-quality development [J]. Statistics and Decision Making,2023,39(07):179-183.

25.Hao X, Li Y, Ren S, et al. The role of digitalization on green economic growth: Does industrial structure optimization and green innovation matter?[J]. Journal of environmental management, 2023, 325: 116504.

26.Wang L, Chen Y, Ramsey T S, et al. Will researching digital technology really empower green development?[J]. Technology in Society, 2021, 66: 101638.

27.Su X, Wang S, Li F. The Impact of Digital Transformation on ESG Performance Based on the Mediating Effect of Dynamic Capabilities[J]. Sustainability, 2023, 15(18): 13506.

28.Wu S, Li Y. A Study on the Impact of Digital Transformation on Corporate ESG Performance: The Mediating Role of Green Innovation[J]. Sustainability, 2023, 15(8): 6568.

29.Wang Yinghuan, Guo Yongzhen. Enterprise digital transformation and ESG performance: Based on empirical evidence of listed enterprises in China [J]. Journal of Finance and Economics, 2019,49(09):94-108.

30.Wang Xu, Zhang Xiaoning, Niu Yuewei. "Data-driven" and "capability curse" : The strategic paradox of enterprise digital transformation under the guidance of green innovation strategy upgrading [J]. Research and Development Management, 2002,34(04):51-65.

31.Lateef A, Omotayo F O. Information audit as an important tool in organizational management: A review of literature[J]. Business Information Review, 2019, 36(1): 15-22.

32.Baraibar-Diez E, Sotorrío L L. The mediating effect of transparency in the relationship between corporate social responsibility and corporate reputation[J]. Revista brasileira de gestão de negócios, 2018, 20: 05-21.

33.Bushman R M, Piotroski J D, Smith A J. What determines corporate transparency?[J]. Journal of accounting research, 2004, 42(2): 207-252.

34.Liu Wei, Li Jianying. Can media buzz effectively reduce the risk of stock price crashes?-- Research on the moderating effect of corporate transparency [J]. Chinese Management Science,2019,27(11):39-49.

35.Manita R, Elommal N, Baudier P, et al. The digital transformation of external audit and its impact on corporate governance[J]. Technological Forecasting and Social Change, 2020, 150: 119751.

36.Chen W, Zhang L, Jiang P, et al. Can digital transformation improve the information environment of the capital market? Evidence from the analysts' prediction behaviour[J]. Accounting & Finance, 2022, 62(2): 2543-2578.

37.Wang S, Esperança J P. Can digital transformation improve market and ESG performance? Evidence from Chinese SMEs[J]. Journal of Cleaner Production, 2023, 419: 137980.

38.Xiao Tusheng, Sun Ruiqi, Yuan Chun, et al. Corporate digital transformation, human capital structure adjustment and labor income share [J]. Management World, 2022, 38 (12): 220-237.

39.Aral S, Weill P. IT assets, organizational capabilities, and firm performance: How resource allocations and organizational differences explain performance variation[J]. Organization science, 2007, 18(5): 763-780.

40.Tambe P, Hitt L M. The productivity of information technology investments: New evidence from IT labor data[J]. Information systems research, 2012, 23(3-part-1): 599-617.

41.Wang Liyan, Zhang Jidong. The Relationship between ERP system implementation and company performance Growth: An empirical Analysis based on the data of Chinese listed companies [J]. Management World, 2007, (03): 116-121+137.

42.Lang M, Lins K V, Maffett M. Transparency, liquidity, and valuation: International evidence on when transparency matters most[J]. Journal of Accounting Research, 2012, 50(3): 729-774.

43.Xin Qing-Quan, Kong Dong-min, Hao Ying. Corporate transparency and stock price volatility [J]. Financial Research, 2014, (10): 193-206.

44.Rudy B C, Johnson A F. Performance, aspirations, and market versus nonmarket investment[J]. Journal of management, 2016, 42(4): 936-959.

45.Li Xi, Zheng Xin, Zhang Jianqi. Will the performance dilemma of manufacturing Enterprises promote innovation? An analysis based on the expansion of expectation gap dimension [J].China Industrial Economy,2018(08):174-192.

46.Fatemi A, Glaum M, Kaiser S. ESG performance and firm value: The moderating role of disclosure[J]. Global finance journal, 2018, 38: 45-64.

47.Nekhili M, Boukadhaba A, Nagati H. The ESG–financial performance relationship: Does the type of employee board representation matter?[J]. Corporate Governance: An International Review, 2021, 29(2): 134-161.

48.Cheng L T W, Sharma P, Broadstock D C. Interactive effects of brand reputation and ESG on green bond issues: A sustainable development perspective[J]. Business strategy and the environment, 2023, 32(1): 570-586.

49. Tang Yaojia, Wang Yu, Tang Chunhui. Digital economy, market structure and innovation performance [J]. China Industrial Economy,2022(10):62-80.

---

## [Decision Letter · Decision Letter 1]

11 Mar 2024

PONE-D-23-39817R1Technology empowerment: Digital transformation and enterprise ESG performance--Evidence from China's manufacturing sectorPLOS ONE

Dear Dr. Li,

Thank you for submitting your manuscript to PLOS ONE. After careful consideration, we feel that it has merit but does not fully meet PLOS ONE’s publication criteria as it currently stands. Therefore, we invite you to submit a revised version of the manuscript that addresses the points raised during the review process. Please submit your revised manuscript by Apr 25 2024 11:59PM. If you will need more time than this to complete your revisions, please reply to this message or contact the journal office at plosone@plos.org. Please include the following items when submitting your revised manuscript:A rebuttal letter that responds to each point raised by the academic editor and reviewer(s). You should upload this letter as a separate file labeled 'Response to Reviewers'.A marked-up copy of your manuscript that highlights changes made to the original version. You should upload this as a separate file labeled 'Revised Manuscript with Track Changes'.An unmarked version of your revised paper without tracked changes. You should upload this as a separate file labeled 'Manuscript'.If applicable, we recommend that you deposit your laboratory protocols in protocols.io to enhance the reproducibility of your results. Protocols.io assigns your protocol its own identifier (DOI) so that it can be cited independently in the future. For instructions see: https://journals.plos.org/plosone/s/submission-guidelines#loc-laboratory-protocols. Additionally, PLOS ONE offers an option for publishing peer-reviewed Lab Protocol articles, which describe protocols hosted on protocols.io. Read more information on sharing protocols at https://plos.org/protocols?utm_medium=editorial-email&utm_source=authorletters&utm_campaign=protocols.

We look forward to receiving your revised manuscript.

Kind regards,

Jianhua Zhu

Academic Editor

PLOS ONE

Journal Requirements:

Reviewers' comments:

Reviewer's Responses to Questions

**Comments to the Author**

1. If the authors have adequately addressed your comments raised in a previous round of review and you feel that this manuscript is now acceptable for publication, you may indicate that here to bypass the “Comments to the Author” section, enter your conflict of interest statement in the “Confidential to Editor” section, and submit your "Accept" recommendation.

Reviewer #1: All comments have been addressed

Reviewer #2: All comments have been addressed

2. Is the manuscript technically sound, and do the data support the conclusions?

Reviewer #1: Yes

Reviewer #2: Yes

3. Has the statistical analysis been performed appropriately and rigorously? 

Reviewer #1: Yes

Reviewer #2: Yes

4. Have the authors made all data underlying the findings in their manuscript fully available?

Reviewer #1: Yes

Reviewer #2: Yes

5. Is the manuscript presented in an intelligible fashion and written in standard English?

Reviewer #1: Yes

Reviewer #2: Yes

6. Review Comments to the Author

Reviewer #1: The author revised and responded to most of the comments, and I am basically satisfied with his revisions and responses. But there is still a more important issue that needs to be solved by the author. details as follows.

The firm fixed effects shall be added in the benchmark model, if not, please explain why.

Reviewer #2: Why there is numberings throughout the manuscript. Remove all the numberings in the manuscript.

Discussion is still lacking its content.

Provide a concise summary of your main findings and interpret the results in the context of your research question or hypothesis. Discuss any unexpected or contradictory findings and offer possible explanations. Compare your findings with previous research in the field. Highlight areas of agreement or divergence between your results and existing literature. Discuss how your findings contribute to, extend, or challenge existing knowledge. Highlight the novelty, significance, or relevance of your findings. Discuss how your study advances knowledge, fills gaps in the literature, or opens up new avenues for research.

Theoretical and practical contributions are essential elements of any study or research endeavour and therefore it should be separated in titles as there is lack of explanation in both parts

Sample of theoretical and practical contributions

Theoretical Contribution

Advancement of Knowledge - A theoretical contribution involves expanding existing knowledge or introducing new theoretical frameworks, models, or concepts.

Integration of Literature - It involves synthesizing and integrating existing literature to develop a deeper understanding of the topic under investigation.

Identification of Gaps - By identifying gaps or inconsistencies in the existing literature, researchers contribute to the advancement of knowledge in their field.

Development of New Perspectives - Researchers may offer new perspectives or interpretations of existing theories or phenomena, enriching the theoretical landscape.

Validation or Refutation - Studies may validate or refute existing theories, leading to a better understanding of the phenomenon being studied.

Practical Contribution

Application of Theory - Practical contributions involve applying theoretical insights to real-world problems or situations, thereby providing actionable solutions.

Development of Tools or Methods - Researchers may develop practical tools, methodologies, or interventions based on theoretical foundations to address practical challenges.

Impact on Practice -Studies that have practical implications can influence decision-making processes, policy development, or organizational practices.

Empirical Evidence - By conducting empirical research, researchers provide evidence to support the practical applicability of theoretical concepts.

Addressing Societal Needs - Practical contributions may address societal needs or concerns, leading to tangible improvements in various domains such as healthcare, education, or technology.

Data analysis was performed using Python and Stata version 16.0 software but it was not stated in the manuscript

Have a title on discussion and discuss thoroughly on whether the objectives achieved or otherwise based on the analysis.

Have a title on conclusion and limitations.

These titles need to be separated.

7. PLOS authors have the option to publish the peer review history of their article (what does this mean?). If published, this will include your full peer review and any attached files.

Reviewer #1: **Yes: **Tenglong Zhong

Reviewer #2: No

---

## [Author Response · Author response to Decision Letter 1]

19 Mar 2024

Dear Editor and reviewers,

We sincerely appreciate your correspondence and the reviewers' insightful comments on our manuscript titled "Technology Empowerment: Digital Transformation and Enterprise ESG Performance - Evidence from China's Manufacturing Sector" (Manuscript ID: PONE-D-23-39817R1).These comments are all valuable and helpful for improving our article .Revisions were extensively made to the manuscript based on feedback from editors and reviewers. In the revised version, changes have been highlighted in blue.

The main changes made in response to comments from editors and reviewers are summarized below:

Responds to the Editor's comments:

1.Please review your reference list to ensure that it is complete and correct. If you have cited papers that have been retracted, please include the rationale for doing so in the manuscript text, or remove these references and replace them with relevant current references. Any changes to the reference list should be mentioned in the rebuttal letter that accompanies your revised manuscript. If you need to cite a retracted article, indicate the article’s retracted status in the References list and also include a citation and full reference for the retraction notice.

Reply:Thank you very much indeed for your comments.According to your suggestion, we have reorganized the references and replaced the original 7, 11, 22 , 24 and 99.The original reference with serial number 100 was deleted.

Responds to the reviewer’s comments:

Reviewer #1:

1.The firm fixed effects shall be added in the benchmark model, if not, please explain why.

Reply:Thank you very much indeed for your comments.In the Benchmark regression analysis, this study builds upon existing research findings and incorporates time fixed effects and industry fixed effects to ensure model stability[1,2]. Additionally, to address potential issues of omitted variables, firm fixed effects, industry fixed effects, and time fixed effects were included in the robustness test. The results remain significant, thus further confirming hypothesis 1 proposed in this study.

The regression results presented in the following table incorporate firm fixed effects, time fixed effects, and industry fixed effects.

 (1)

VARIABLES ESG

Digital 0.542***

 (0.148)

Size 1.895***

 (0.189)

Lev -3.331***

 (0.718)

Grow -0.336*

 (0.192)

Cashflow 1.508

 (1.132)

Indire 0.953

 (1.765)

Listage 2.114

 (1.404)

Top1 1.862

 (1.145)

Constant -23.092***

 (5.589)

Observations 5,962

R-squared 0.848

Robust standard errors in parentheses

*** p<0.01, ** p<0.05, * p<0.1

Reviewer #2:

1.Why there is numberings throughout the manuscript. Remove all the numberings in the manuscript.

Reply:Thank you very much indeed for your comments.We have removed all numbers from the article.

2.Discussion is still lacking its content.

Provide a concise summary of your main findings and interpret the results in the context of your research question or hypothesis. Discuss any unexpected or contradictory findings and offer possible explanations. Compare your findings with previous research in the field. Highlight areas of agreement or divergence between your results and existing literature. Discuss how your findings contribute to, extend, or challenge existing knowledge. Highlight the novelty, significance, or relevance of your findings. Discuss how your study advances knowledge, fills gaps in the literature, or opens up new avenues for research.

Reply:Thank you very much indeed for your comments.According to your suggestions, we have made detailed revisions to the discussion section. The specific modifications are as follows:Firstly, the empirical analysis results confirm the hypothesis (H1) proposed in this study. This finding aligns with existing research and further substantiates that digital transformation not only positively impacts financial performance but also serves as an internal driver for enhancing ESG performance within enterprises .

 Furthermore, this study confirms the proposition H2. Existing literature predominantly examines the relationship between digital transformation and ESG performance through the lenses of total factor productivity and dynamic capability, neglecting the role of corporate transparency in this context. On one hand, the application of digital technology brings about technological advantages that generate a "governance effect," enhancing internal governance capabilities by increasing shareholder participation in decision-making and curbing managerial discretion. On the other hand, digital transformation yields a "spotlight effect" that amplifies market attention towards enterprises and facilitates greater interaction frequency of internal and external information, thereby promoting environmental and social investments among manufacturing firms to enhance their ESG performance. This research finding expands upon existing knowledge regarding the mechanisms linking digital transformation with enterprise ESG performance while contributing to non-economic value research within the domain of digital transformation.

This study also investigates whether the relationship between digital transformation and ESG performance is influenced by the performance expectation gap, and the empirical findings confirm the hypothesis H3b proposed in this study. The underlying reason is that during a performance expectation gap, enterprises face increased strategic risks and heightened internal and external pressures on management. Pursuing strategic reforms at such times may not yield immediate turnaround results but can potentially lead to organizational difficulties. Consequently, the performance expectation gap tends to foster more conservative strategic decision-making by management, thereby limiting the extent to which digital transformation can promote ESG performance. This conclusion underscores the significance of performance feedback in understanding the intrinsic connection between digital transformation and ESG performance within enterprises while addressing existing research limitations.

3.Data analysis was performed using Python and Stata version 16.0 software but it was not stated in the manuscript

Reply:Thank you very much indeed for your comments.According to your suggestion, we have included a description of the software used for data analysis in the data sources section.

4.Have a title on discussion and discuss thoroughly on whether the objectives achieved or otherwise based on the analysis.

Have a title on conclusion and limitations.

These titles need to be separated.

Reply:Thank you very much indeed for your comments. According to your suggestions, we have revised the conclusions, discussions, and limitations of the article. The specific changes have been highlighted in blue in the revised manuscript.

According to the editor and reviewers’ comments, we have made extensive modifications to our manuscript and supplemented extra data to make our results convincing. Thank you again for your positive comments and valuable suggestions to improve the quality of our manuscript.If there are any other modifications we could make, we would like very much to modify them and we really appreciate your help.

Sincerely,

Longji Li

1.Apergis N, Poufinas T, Antonopoulos A. ESG scores and cost of debt[J]. Energy Economics, 2022, 112: 106186.

2.Liu Xiangqiang, Yang Qingqing, Hu Jun.ESG rating divergence and Stock price Synchronization [J]. China Soft Science,2023(08):108-120.

---

## [Decision Letter · Decision Letter 2]

27 Mar 2024

Technology empowerment: Digital transformation and enterprise ESG performance--Evidence from China's manufacturing sector

PONE-D-23-39817R2

Dear Dr. Li,

We’re pleased to inform you that your manuscript has been judged scientifically suitable for publication and will be formally accepted for publication once it meets all outstanding technical requirements.

Kind regards,

Jianhua Zhu

Academic Editor

PLOS ONE

Additional Editor Comments (optional):

Reviewers' comments:

Reviewer's Responses to Questions

**Comments to the Author**

1. If the authors have adequately addressed your comments raised in a previous round of review and you feel that this manuscript is now acceptable for publication, you may indicate that here to bypass the “Comments to the Author” section, enter your conflict of interest statement in the “Confidential to Editor” section, and submit your "Accept" recommendation.

Reviewer #1: All comments have been addressed

2. Is the manuscript technically sound, and do the data support the conclusions?

Reviewer #1: Yes

3. Has the statistical analysis been performed appropriately and rigorously? 

Reviewer #1: Yes

4. Have the authors made all data underlying the findings in their manuscript fully available?

Reviewer #1: Yes

5. Is the manuscript presented in an intelligible fashion and written in standard English?

Reviewer #1: Yes

6. Review Comments to the Author

Reviewer #1: The author carefully revised and responded based on the review comments. I am very satisfied and have no other comments.

7. PLOS authors have the option to publish the peer review history of their article (what does this mean?). If published, this will include your full peer review and any attached files.

Reviewer #1: **Yes: **Tenglong Zhong

---

## [Editor Report · Acceptance letter]

1 Apr 2024

PONE-D-23-39817R2 

PLOS ONE

Dear Dr. Li, 

I'm pleased to inform you that your manuscript has been deemed suitable for publication in PLOS ONE. Congratulations! Your manuscript is now being handed over to our production team.

Kind regards, 

on behalf of

Dr. Jianhua Zhu 

Academic Editor

PLOS ONE